# A Simple and Effective Pruning Approach for Large Language Models

**Mingjie Sun**[1][*]     **Zhuang Liu**[2][*]     **Anna Bair**[1]     **J. Zico Kolter**[1,3]
[1]Carnegie Mellon University     [2]Meta AI Research     [3]Bosch Center for AI

## Abstract

As their size increases, Large Languages Models (LLMs) are natural candidates for network pruning methods: approaches that drop a subset of network weights while striving to preserve performance. Existing methods, however, require either retraining, which is rarely affordable for billion-scale LLMs, or solving a weight reconstruction problem reliant on second-order information, which may also be computationally expensive. In this paper, we introduce a novel, straightforward yet effective pruning method, termed Wanda (Pruning by **W**eights **and a**ctivations), designed to induce sparsity in pretrained LLMs. Motivated by the recent observation of emergent large magnitude features in LLMs, our approach prunes weights with the smallest magnitudes multiplied by the corresponding input activations, on a *per-output* basis. Notably, Wanda requires *no* retraining or weight update, and the pruned LLM can be used *as is*. We conduct a thorough evaluation of our method Wanda on LLaMA and LLaMA-2 across various language benchmarks. Wanda significantly outperforms the established baseline of magnitude pruning and performs competitively against recent method involving intensive weight update. Code is available at https://github.com/locuslab/wanda.

## 1 Introduction

Large language models (Brown et al., 2020; OpenAI, 2023) have recently reshaped the field of NLP with their remarkable performance across a range of complex language benchmarks (Bommarito & Katz, 2022; Wei et al., 2022a; Bubeck et al., 2023). However, these models, with their billions of parameters, usually require significant computational resources. To democratize LLMs, considerable efforts have been taken to mitigate their high computational cost. Many of the notable advancements to date have centered on model quantization, a process where parameters are quantized into lower bit-level representations. The fast pace of LLM quantization research (Dettmers et al., 2022; Frantar et al., 2023a; Xiao et al., 2023; Ahmadian et al., 2023) has led to substantial resource savings for these models (Sheng et al., 2023; Lin et al., 2023).

Network pruning (LeCun et al., 1989; Hassibi et al., 1993; Han et al., 2015), on the other hand, shrinks network sizes by removing specific weights from the model – essentially setting them to zero. Along with quantization, it is often considered another popular approach for compressing neural networks. However, it has received relatively little focus in compressing LLMs. This seems to contradict the trend of model compression in the pre-LLM era, where both approaches have received large amounts of research effort. A quick review of existing pruning methods reveals a possible reason: they typically require retraining (Liu et al., 2019; Blalock et al., 2020), training from random initializations (Zhu & Gupta, 2017; Louizos et al., 2018; Gale et al., 2019) or even an extensive iterative process (Frankle & Michael, 2019; Renda et al., 2020). The sheer amount of computational resources required by LLMs limits these methods. A recent LLM pruning approach, SparseGPT (Frantar & Alistarh, 2023), does not require traditional retraining, but still demands a computationally intensive weight update process.

The argument concerning the need for retraining and weight update does not fully capture the challenges of pruning LLMs. One might reasonably expect to obtain a fairly high-performing initialization point for retraining using existing popular pruning methods. However, a recent study (Frantar &

---

[*]Equal contribution. Correspondence to mingjies@cs.cmu.edu and zhuangl@meta.com.

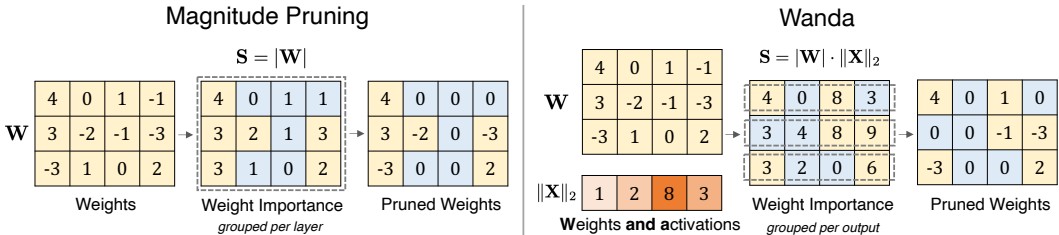

Figure 1: Illustration of our proposed method Wanda (Pruning by **W**eights **and a**ctivations), compared with the magnitude pruning approach. Given a weight matrix $\mathbf{W}$ and input feature activations $\mathbf{X}$, we compute the weight importance as the product between the weight magnitude and the norm of the corrsponding input activations ($|\mathbf{W}| \cdot \|\mathbf{X}\|_2$). Weight importance scores are compared on a *per-output* basis (within each row in $\mathbf{W}$), rather than globally across the entire matrix.

Alistarh, 2023) finds that magnitude pruning (Han et al., 2015), a well-established pruning approach, fails dramatically on LLMs even with relatively low levels of sparsity. Considering the past success of magnitude pruning on smaller networks, this result suggests that LLMs, despite having 100 to 1000 times more parameters, are substantially more difficult to prune directly.

In this work, we address this challenge by introducing a straightforward and effective approach, termed Wanda (Pruning by **W**eights **and a**ctivations). This technique successfully prunes LLMs to high degrees of sparsity without *any* need for modifying the remaining weights. We are motivated by an observation from a recent study (Dettmers et al., 2022), where a small subset of hidden state features are exceptionally large in magnitude, a property unique to LLMs. We find that augmenting the standard weight magnitude pruning metric with the input activations, is surprisingly effective as a measure for evaluating the weight importance. Specifically, we introduce a novel pruning metric, where each weight is evaluated by the product of its magnitude and the norm of the corresponding input activations, estimated using a small set of calibration data. Our method uses this metric to induce sparsity in pretrained LLMs by comparing weights locally within each output of linear layers and removing lower priority weights. Our approach is computationally efficient, able to be executed in a single forward pass, and requires minimal memory overhead.

We empirically evaluate Wanda on the widely adopted LLaMA (Touvron et al., 2023a) and LLaMA-2 (Touvron et al., 2023b) model families. Our results demonstrate Wanda can find efficient sparse networks from pretrained LLMs, without any retraining or weight update. Our approach Wanda outperforms the standard magnitude pruning by a large margin and also competes favorably with the prior best LLM pruning method (Frantar & Alistarh, 2023), while requiring a lower computational cost. We hope our work serves as a baseline for future work in this area, and encourages further exploration in understanding sparsity in LLMs.

## 2    PRELIMINARIES

**Magnitude Pruning** (Han et al., 2015) is a standard pruning technique to induce sparsity in neural networks. It removes individual weights based on their magnitudes, where weights with magnitudes below a certain threshold are removed. In practice, this threshold is typically determined by comparing weights locally within each layer or globally across the whole network. Despite its simplicity, magnitude pruning has been used to find extremely sparse networks (Frankle & Michael, 2019) and now stands out as a strong baseline approach (Blalock et al., 2020) for neural network sparsification.

**Emergent Large Magnitude Features** have been observed in Transformer-based large language models. Dettmers et al. (2022) discover that once LLMs reach a certain scale (in practice, around 6B parameters), a small set of hidden state features emerges with significantly larger magnitudes than the remaining ones. These outlier features exhibit several intriguing characteristics. First, they have very large magnitudes, about 100 times larger than typical hidden state values. Second, they are usually sparse and exist in certain feature dimensions. Finally, these outlier features are essential for the predictive capability of LLMs: zeroing out these features at inference time results in significant degradation of language modeling performance.

## 3 WANDA: PRUNING BY WEIGHTS AND ACTIVATIONS

In this section, we motivate and describe our pruning method, Wanda (Pruning by **W**eights **and** **a**ctivations), which consists of two simple but essential components. First, we propose a novel pruning metric that incorporates both weights and input activations into the computation of weight importance. Second, we compare weights on a *per-output* basis instead of across the whole layer, which we find is crucial for pruning LLMs effectively. An overview of Wanda is shown in Figure 1.

**A Motivating Example.** Consider a neuron with two inputs and corresponding weights: $\mathbf{y} = \mathbf{w}_1\mathbf{x}_1 + \mathbf{w}_2\mathbf{x}_2$, where $|\mathbf{w}_1| \leq |\mathbf{w}_2|$. Now suppose the goal is to select one weight for removal while incurring less change on the output. The standard approach of magnitude pruning would always remove weight $\mathbf{w}_1$, which may be a good strategy if input features $\mathbf{x}_1$ and $\mathbf{x}_2$ have similar magnitudes. However, as recently observed in LLMs (Dettmers et al., 2022), the two input features can differ significantly in scale. For instance, it is possible that $|\mathbf{x}_1| \gg |\mathbf{x}_2|$, and as a result, $|\mathbf{w}_1\mathbf{x}_1| \gg |\mathbf{w}_2\mathbf{x}_2|$. In this case, we should remove weight $\mathbf{w}_2$ instead, because this removal clearly exerts a smaller influence on the neuron output $\mathbf{y}$ than removing weight $\mathbf{w}_1$.

This motivating example with the simplest linear layer hints at a major limitation of magnitude pruning: it does not take into account input activations, which could play an equally important role as weight magnitudes in determining the neuron output. For pruning LLMs, this is especially critical considering the emergent large magnitude features found within them. Thus, as the first part of our method, we propose a pruning metric designed explicitly for LLMs to handle such a limitation, while also maintaining the simplicity of magnitude pruning.

**Pruning Metric.** Consider a linear layer with weight $\mathbf{W}$ of shape $(C_{\text{out}}, C_{\text{in}})$. For language models, this linear layer takes in input activations $\mathbf{X}$ with a shape of $(N \times L, C_{\text{in}})$, where $N$ and $L$ are batch and sequence dimensions respectively. For each individual weight, we propose to evaluate its importance by the product of its magnitude and the corresponding input feature norm. Specifically, the score for the current weight $\mathbf{W}_{ij}$ is defined by:

$$\mathbf{S}_{ij} = |\mathbf{W}_{ij}| \cdot \|\mathbf{X}_j\|_2 \tag{1}$$

where $|\cdot|$ represents the absolute value operator, $\|\mathbf{X}_j\|_2$ evaluates the $\ell_2$ norm of $j$th features aggregated across $N \times L$ different tokens, and the final score is computed by the product of these two scalar values. We find that $\ell_2$ norm tends to work better than other norm functions (e.g., $\ell_1$ and $\ell_\infty$) in measuring activation magnitudes. This is possibly because $\ell_2$ norm is generally a smoother metric.

This metric is interesting in several aspects. First, when the input channel of the considered weight has large magnitude features, the weight itself tends to be assigned a larger importance score even if it has a low magnitude. This tackles the problem we encounter in the motivating example. The effect can be seen in Figure 1, where weights corresponding to the large magnitude feature are more likely to be preserved with Wanda. Second, its computation is straightforward. Once we obtain the norm vector of input feature activations, the weight importance can be calculated using an element-wise dot product. Last, we find empirically that this metric is robust and can be easily estimated using a modest number of calibration samples, without access to the original training data.

**Comparison Group.** Generally, in a pruning method, each weight is first assigned an importance score, such as the pruning metric we discussed above. These weights are then grouped into *comparison groups* where weights within each group are compared against one another. Within each comparison group, weights with lower importance scores are pruned. Most previous pruning methods default to comparing weights locally within each layer or globally across the whole network.

While layer-wise and whole-network comparisons have been the popular options, we find that pruning LLMs could benefit from a more localized grouping. In our method, we compare and remove weights on a *per-output* basis (per row in Figure 1), where weight importance scores are compared locally within each output neuron. Specifically, for a weight $\mathbf{W}_{ij}$ that connects input $j$ to output $i$ inside the linear layer, we define the *comparison group* for this weight as all weights connecting to output $i$:

$$\mathbf{G}_{ij} = \{\mathbf{W}_{uv} \,|\, u = i\} \tag{2}$$

Under this comparison group, for a pre-defined sparsity ratio $s\%$, we eliminate $s\%$ of the weights connected to *each output*. This practice may seem counter-intuitive, since we are basically pruning under a stricter sparsity pattern. However, we find that it is *consistently better* than layer-wise pruning for LLMs. Notably, this holds true not only for our proposed pruning metric (Equation 1) but also

the standard magnitude metric. This shows that maintaining a balanced pruning ratio across output features is important for pruning LLMs effectively.

To see if the superiority of pruning *per output* over *per layer* holds true in general, we evaluate on image classifiers. However, we do not observe similar trend in image classification models, suggesting that our observations regarding pruning *per output* might be unique to LLMs. We hope this intriguing observation encourages practitioners to be more cautious in choosing the comparison group.

**Procedure.** Wanda can be implemented and integrated seamlessly within a *single* forward pass of the LLM model, where feature norm statistics $\|\mathbf{X}_j\|_2$ are estimated with a set of calibration data. We provide the PyTorch code of our approach in Algorithm 1. Given a pretrained LLM, we compute our pruning metric from the initial to the final layers of the network. After pruning a preceding layer, the current layer receives the updated input activations, obtained on the pruned weights of the previous layer. Then the pruning metrics are computed. A recent method for pruning

---

**Algorithm 1** PyTorch code for Wanda

```python
# W: weight matrix (C_out, C_in);
# X: input matrix (N * L, C_in);
# s: desired sparsity, between 0 and 1;

def prune(W, X, s):
  metric = W.abs() * X.norm(p=2, dim=0)

  _, sorted_idx = torch.sort(metric, dim=1)
  pruned_idx = sorted_idx[:,:int(C_in * s)]
  W.scatter_(dim=1, index=pruned_idx, src=0)

  return W
```

---

LLMs, SparseGPT (Frantar & Alistarh, 2023), requires a sophisticated weight update procedure in an iterative pruning process, while Wanda does not induce any additional weight update.

**Structured N:M Sparsity.** While Wanda so far has been developed for unstructured sparsity, it can be easily extended to structured N:M sparsity (Mishra et al., 2021). Structured N:M sparsity requires that at most N out of every M contiguous weights to be non-zero. It can leverage NVIDIA's sparse tensor cores to accelerate matrix multiplication in practice. Wanda can be naturally extended to structured N:M sparsity, where we compare weights using the same metric among every M consecutive weights, for all weights connected to an output.

**Remark.** We discuss the connection between Wanda and a few existing works. SparseGPT formalizes the problem of pruning LLMs by solving a local layer-wise reconstruction problem, where their pruning metric and weight update procedure is inspired from Optimal Brain Surgeon (OBS) (Hassibi et al., 1993). The pruning metric in SparseGPT is:

$$\mathbf{S}_{ij} = \left[ |\mathbf{W}|^2 / \mathrm{diag}\big( (\mathbf{X}^T\mathbf{X} + \lambda\mathbf{I})^{-1} \big) \right]_{ij} \tag{3}$$

Here $\mathbf{X}^T\mathbf{X} + \lambda\mathbf{I}$ in the denominator is the Hessian $\mathbf{H}$ for the layer-wise reconstruction problem and $\lambda$ is the Hessian dampening factor to avoid the collapse of inverse computation. With careful inspection, we observe that our metric in Equation 1 is similar to the above when $\lambda$ is 0 and only the diagonal elements of the Hessian matrix $\mathbf{X}^T\mathbf{X} + \lambda\mathbf{I}$ are retained. Starting from the pruning metric in Equation 3, we show the exact reduction steps and corresponding reduction conditions as follows:

$$\mathbf{S}_{ij} \overset{\lambda=0}{=} \left[ |\mathbf{W}|^2 / \mathrm{diag}\Big( (\mathbf{X}^T\mathbf{X})^{-1} \Big) \right]_{ij} \overset{\text{diagonal}}{\underset{\text{approx.}}{=}} \left[ |\mathbf{W}|^2 / \Big( \mathrm{diag}(\mathbf{X}^T\mathbf{X}) \Big)^{-1} \right]_{ij} = \big( |\mathbf{W}_{ij}| \cdot \|\mathbf{X}_j\|_2 \big)^2$$

In the last reduction step, the $j$th diagonal of $\mathbf{X}^T\mathbf{X}$ is $\|\mathbf{X}_j\|_2^2$, and thus the denominator can be simplified to $(\|\mathbf{X}_j\|_2^2)^{-1}$. This simplification substantially reduces the required computation of weight importance, eliminating the need for computing any matrix inverses.

In the 1980s, LeCun et al. (1989) have set up a pioneering framework for neural network pruning named Optimal Brain Damage (OBD). It uses second-order information without off-diagonal elements in Hessians for faster approximation. Later, Optimal Brain Surgeon (OBS) develops upon OBD partly by taking into account the off-diagonal elements. Wanda can be seen as a renaissance of OBD – it may be viewed as applying a process similar to OBD to each neuron, with *local* output reconstruction as the objective function, whereas the original OBD uses the *global* training objective. This is analogous to the relationship between SparseGPT and OBS.

A comparison of LLM pruning methods can be found in Table 1. Computing the pruning metric of Wanda has a reduced time complexity compared to SparseGPT, because it does not involve inverse computation. Overall, our method Wanda (Pruning by **W**eights **and a**ctivations) has several attractive properties as an approach for pruning LLMs:

| Method | Weight Update | Calibration Data | Pruning Metric $\mathbf{S}_{ij}$ | Complexity |
|---|---|---|---|---|
| Magnitude | ✗ | ✗ | $\|\mathbf{W}_{ij}\|$ | $O(1)$ |
| SparseGPT | ✓ | ✓ | $\left[\|\mathbf{W}\|^2/\text{diag}\left[(\mathbf{XX}^T + \lambda\mathbf{I})^{-1}\right]\right]_{ij}$ | $O(d_{\text{hidden}}^3)$ |
| Wanda | ✗ | ✓ | $\|\mathbf{W}_{ij}\| \cdot \|\mathbf{X}_j\|_2$ | $O(d_{\text{hidden}}^2)$ |

Table 1: Comparison of Wanda with existing pruning algorithms on LLMs.

1. It maintains the simplicity of magnitude pruning in the pre-LLM era, requiring no gradient computation via back-propagation or any second-order Hessian inverses, but is also highly effective in discovering sparse networks in pretrained LLMs.

2. Wanda can be done with a single forward pass of the LLM. At each layer, the pruned weights can be decided in one shot without an iterative procedure. In practice, computing the pruning metric of Wanda can be 300 times faster in pruning LLMs compared with SparseGPT.

3. Unlike SparseGPT, our approach entails no weight update on pruned networks, suggesting that LLMs have effective sparse sub-networks that are *exact*, instead of them merely existing in the neighborhood of the original weights.

## 4 EXPERIMENTS

**Models and Evaluation.** We evaluate Wanda on the two most widely adopted LLM model families: LLaMA 7B/13B/30B/65B (Touvron et al., 2023a) and LLaMA-2 7B/13B/70B (Touvron et al., 2023b) (LLaMA-2 34B is not released). Results for prior LLM families can be found in Appendix B. We measure the performance of pruned models on zero-shot tasks and language modeling. For zero-shot evaluation, we use seven tasks from EleutherAI LM Harness (Gao et al., 2021). Following previous works on LLM compression (Xiao et al., 2023; Frantar & Alistarh, 2023), we evaluate the perplexity on the held-out WikiText (Merity et al., 2016) validation set.

**Baselines.** We compare Wanda with two prior pruning approaches. Magnitude pruning (Han et al., 2015) is a simple and strong baseline in which weights are discarded based on their magnitudes. SparseGPT (Frantar & Alistarh, 2023) is a second-order pruning method for LLMs, based on solving a layer-wise reconstruction problem. In Appendix C, we compare with additional pruning methods.

Both Wanda and SparseGPT require calibration data to estimate input statistics (see Table 1). To control this variable factor, we use the *exact same* set of calibration data as SparseGPT, which consists of 128 sequences with context length size sampled from C4 training set (Raffel et al., 2020). In Appendix D.1, we provide additional analysis on the number of calibration samples.

**Sparsity.** For all pruning methods, we focus on pruning the linear layers (skipping the first embedding layer and the final classification head), which account for around 99% of the total LLM parameters. We impose a uniform sparsity for all linear layers. We evaluate three types of sparsity: unstructured sparsity, structured 4:8 and 2:4 sparsities. The magnitude pruning baseline is extended to structured N:M sparsity in a similar spirit to our method, as described in the previous section.

### 4.1 ZERO-SHOT TASKS

**Comparison with Baselines.** In Table 2, we show the mean zero-shot accuracies on 7 zero-shot tasks of pruned LLaMA and LLaMA-2 models. We refer the reader to Appendix D for task-wise performance. Across both unstructured and structured sparsities, Wanda outperforms the well-established magnitude pruning approach by a large margin, while also rivals with the previous best approach SparseGPT. Given that no fine-tuning takes place, there is a noticeable gap between sparse pruned LLMs and the original dense LLMs. However, as the model size increases, this accuracy gap diminishes. Remarkably, unstructured 50% sparse LLaMA-65B and LLaMA-2-70B is able to match the zero-shot accuracies of their dense counterparts.

**Large Sparse *vs.* Small Dense.** It might be of interest to some readers on the comparison between large sparse LLMs and small dense LLMs with similar parameter counts. For zero-shot performance, we find the trend differs across the types of sparsity. For unstructured sparsity, large sparse LLMs are often better than small dense LLMs on zero-shot performance: unstructured 50% sparse LLaMA-65B (66.67%) outperforms dense LLaMA-30B (65.38%); unstructured 50% sparse LLaMA-2-13B (60.83%) outperforms dense LLaMA-7B (59.71%). Intriguingly, this gap is much larger for few-shot tasks (see Appendix D). For structured sparsity, the trend is reversed: without any fine-tuning, large sparse LLMs have worse zero-shot performance than small dense LLMs in general.

| Method | Weight Update | Sparsity | LLaMA | | | | LLaMA-2 | | |
|---|---|---|---|---|---|---|---|---|---|
| | | | 7B | 13B | 30B | 65B | 7B | 13B | 70B |
| Dense | - | 0% | 59.99 | 62.59 | 65.38 | 66.97 | 59.71 | 63.03 | 67.08 |
| Magnitude | ✗ | 50% | 46.94 | 47.61 | 53.83 | 62.74 | 51.14 | 52.85 | 60.93 |
| SparseGPT | ✓ | 50% | **54.94** | 58.61 | 63.09 | 66.30 | **56.24** | 60.72 | **67.28** |
| Wanda | ✗ | 50% | 54.21 | **59.33** | **63.60** | **66.67** | 56.24 | **60.83** | 67.03 |
| Magnitude | ✗ | 4:8 | 46.03 | 50.53 | 53.53 | 62.17 | 50.64 | 52.81 | 60.28 |
| SparseGPT | ✓ | 4:8 | **52.80** | 55.99 | 60.79 | 64.87 | **53.80** | **59.15** | 65.84 |
| Wanda | ✗ | 4:8 | 52.76 | **56.09** | **61.00** | **64.97** | 52.49 | 58.75 | **66.06** |
| Magnitude | ✗ | 2:4 | 44.73 | 48.00 | 53.16 | 61.28 | 45.58 | 49.89 | 59.95 |
| SparseGPT | ✓ | 2:4 | **50.60** | **53.22** | 58.91 | 62.57 | **50.94** | 54.86 | 63.89 |
| Wanda | ✗ | 2:4 | 48.53 | 52.30 | **59.21** | **62.84** | 48.75 | **55.03** | **64.14** |

Table 2: Mean zero-shot accuracies (%) of pruned LLaMA and LLaMA-2 models. Wanda performs competitively against prior best method SparseGPT, without introducing any weight update.

| Method | Weight Update | Sparsity | LLaMA | | | | LLaMA-2 | | |
|---|---|---|---|---|---|---|---|---|---|
| | | | 7B | 13B | 30B | 65B | 7B | 13B | 70B |
| Dense | - | 0% | 5.68 | 5.09 | 4.77 | 3.56 | 5.12 | 4.57 | 3.12 |
| Magnitude | ✗ | 50% | 17.29 | 20.21 | 7.54 | 5.90 | 14.89 | 6.37 | 4.98 |
| SparseGPT | ✓ | 50% | **7.22** | 6.21 | 5.31 | **4.57** | 6.51 | 5.63 | **3.98** |
| Wanda | ✗ | 50% | 7.26 | **6.15** | **5.24** | **4.57** | **6.42** | **5.56** | **3.98** |
| Magnitude | ✗ | 4:8 | 16.84 | 13.84 | 7.62 | 6.36 | 16.48 | 6.76 | 5.54 |
| SparseGPT | ✓ | 4:8 | 8.61 | **7.40** | 6.17 | 5.38 | 8.12 | 6.60 | 4.59 |
| Wanda | ✗ | 4:8 | **8.57** | **7.40** | **5.97** | **5.30** | **7.97** | **6.55** | **4.47** |
| Magnitude | ✗ | 2:4 | 42.13 | 18.37 | 9.10 | 7.11 | 54.59 | 8.33 | 6.33 |
| SparseGPT | ✓ | 2:4 | **11.00** | **9.11** | 7.16 | 6.28 | **10.17** | 8.32 | 5.40 |
| Wanda | ✗ | 2:4 | 11.53 | 9.58 | **6.90** | **6.25** | 11.02 | **8.27** | **5.16** |

Table 3: WikiText perplexity of pruned LLaMA and LLaMA-2 models. Wanda performs competitively against prior best method SparseGPT, without introducing any weight update.

## 4.2 LANGUAGE MODELING

In Table 3, we report the perplexity of pruned LLaMA and LLaMA-2 models. For robustness analysis under random sampling of the calibration data, see Appendix D.

Without any weight update, Wanda outperforms the established pruning approach of magnitude pruning by a large margin. For instance, for LLaMA-7B, Wanda is able to find sparse networks with a perplexity of 7.26, significantly better than the magnitude pruning baseline 17.29. This result suggests that exact and effective sparse sub-networks exist for LLMs. For unstructured 50% sparsity, Wanda performs on par with the prior best approach SparseGPT. We provide results for higher sparsity levels (60% and 80%) in Appendix D. The comparison between Wanda and SparseGPT is mixed for structured sparsity. On smaller models (e.g., 7B), SparseGPT outperforms Wanda on 2:4 sparsity. Wanda is more favorable for larger models, e.g., LLaMA-30B (2:4 and 4:8) and LLaMA-2-70B (2:4).

## 4.3 SPEEDUP

**Pruning Speed.** The theoretical computational complexity of Wanda is lower than SparseGPT (Table 1). Here we compare their empirical pruning speed. Specifically, we measure the accumulated time for computing the pruning metric at each layer (excluding the forward pass process shared by both methods) on NVIDIA A6000 GPUs. Results are shown in Table 4. Wanda incurs negligible time overhead relative to SparseGPT. The fast speed of Wanda is particularly useful when pruning needs to be performed on a real-time basis, e.g., training sparse models from scratch (Evci et al., 2020) and finding the optimal sparsity (Jin et al., 2022).

**Inference Speed.** We evaluate the inference speedup for structured 2:4 sparsity on NVIDIA A6000 GPUs. Following the evaluation setup of Frantar & Alistarh (2023), we measure the latency of matrix multiplication in linear layers. We perform simulation analysis using the high-performance GEMM kernel in NVIDIA CUTLASS library. Results for LLaMA-65B (batch size of 1) can be found in Table 5. Structured 2:4 sparsity is able to bring notable inference speedup (around 1.6×) for linear layers in LLMs. For end to end latency, we observe a speedup of 1.24× on LLaMA-7B (251ms as compared to 312ms). Last, we emphasize that the inference speedup is not unique to our pruning method but is delivered by the inherent power of sparsity for speeding up computation.

| | LLaMA | | | |
|---|---|---|---|---|
| Method | 7B | 13B | 30B | 65B |
| SparseGPT | 203.1 | 339.0 | 810.3 | 1353.4 |
| Wanda | **0.54** | **0.91** | **2.9** | **5.6** |

Table 4: Computing the pruning metric of Wanda can be much faster (seconds) than SparseGPT.

| LLaMA Layer | Dense | 2:4 | Speedup |
|---|---|---|---|
| `q/k/v/o_proj` | 3.49 | 2.14 | 1.63× |
| `up/gate_proj` | 9.82 | 6.10 | 1.61× |
| `down_proj` | 9.92 | 6.45 | 1.54× |

Table 5: Speedup of matrix multiplication (ms) in LLaMA-65B, for structured 2:4 sparsity.

## 5 ANALYSIS

We study several aspects of Wanda to better understand its effectiveness in pruning LLMs. We use the LLaMA-7B model and prune to unstructured 50% sparsity, unless otherwise specified.

**Fine-tuning.** We study how fine-tuning could recover the performance drop of pruned LLMs, as observed in the previous section. We investigate two strategies for fine-tuning LLMs: LoRA (Hu et al., 2021) fine-tuning and full parameter dense fine-tuning. Fine-tuning is conducted on C4 training dataset and the objective is the pre-training auto-regressive loss. The pruned mask is kept fixed during fine-tuning. We fine-tune pruned LLaMA-7B with all three types of sparsities: unstructured 50%, structured 4:8 and 2:4. Table 6 summarizes the results for mean zero-shot accuracies and perplexity after fine-tuning Wanda pruned LLaMA-7B models. See Appendix D for task-wise performance.

*LoRA Fine-tuning.* We enforce a limited computational budget (1 GPU and 12 hours). The low rank ($r = 8$) adapter is applied on the query and value projection matrices in attention layers. For LLaMA-7B, LoRA introduces only around 0.06% additional parameters, leaving the total sparsity level still around 50%. With LoRA fine-tuning, we are able to restore the performance of pruned LLMs by a non-trivial amount. One notable instance is that LoRA fine-tuning improves the zero-shot performance of structured 2:4 sparse LLaMA-7B from 48.53% to 54.46%, outperforming the original unstrucutred 50% sparse LLaMA-7B (54.21%).

| Evaluation | Dense | Fine-tuning | 50% | 4:8 | 2:4 |
|---|---|---|---|---|---|
| Zero-Shot | 59.99 | ✗ | 54.21 | 52.76 | 48.53 |
| | | LoRA | 56.53 | 54.87 | 54.46 |
| | | Full | **58.15** | **56.65** | **56.19** |
| Perplexity | 5.68 | ✗ | 7.26 | 8.57 | 11.53 |
| | | LoRA | 6.84 | 7.29 | 8.24 |
| | | Full | **5.98** | **6.63** | **7.02** |

Table 6: Fine-tuning can mitigate the gap to dense LLM.

*Full Parameter Fine-tuning.* We conduct full parameter dense fine-tuning. We enforce a limited computational budget (4 GPU and 3 days). Compared to LoRA fine-tuning, full parameter dense fine-tuning is able to mitigate the gap between pruned LLMs and dense LLMs even further. For unstructured 50% sparsity, full parameter fine-tuning could improve pruned LLaMA-7B from 54.21% to 58.15% in terms of zero-shot accuracy, close to that of dense LLaMA-7B (59.99%).

**Pruning Configuration.** Wanda differs from previous methods in both the pruning metric and the comparison group. We conduct ablation experiments to better understand their impact. The three pruning metrics can be found in Table 1. SparseGPT adopts a local comparison group inside a layer, where weights connected to 128 consecutive *input* channels form a group. Wanda groups weights connected with a single *output* channel. Therefore, we ablate two blocksize options (128 and 1) and the input/output choice. For simplicity, we use (input/output, blocksize) to denote each local comparison group, e.g., (input, 1). For this experiment, we do not perform the weight update procedure in SparseGPT to focus on the pruning configuration.

| Pruning Metric | Comparison Group | | | | |
|---|---|---|---|---|---|
| | layer | (input, 1) | (input, 128) | (output, 1) | (output, 128) |
| Magnitude: $\|\mathbf{W}_{ij}\|$ | 17.29 | **8.86** | 16.82 | 13.41 | 17.47 |
| SparseGPT: $\left[\|\mathbf{W}\|^2/\text{diag}(\mathbf{H}^{-1})\right]_{ij}$ | 7.91 | 8.86 | 8.02 | **7.41** | 7.74 |
| Wanda: $\|\mathbf{W}_{ij}\| \cdot \|\mathbf{X}_j\|$ | 7.95 | 8.86 | 8.12 | **7.26** | 7.71 |

Table 7: Ablation on the pruning configuration. **Bold** results denote the best comparison group for each pruning metric. Underscored results indicate the default pruning configuration of each method.

The results are shown in Table 7. We refer the reader to Appendix A for analysis on image classifiers and Appendix D for analysis on previous LLMs. The default pruning configuration of Wanda delivers the best pruned model (perplexity 7.26). Interestingly, for the magnitude metric, comparing weights of the same input neuron (input, 1) yields a perplexity of 8.86, significantly better than other grouping options. Three methods also produce equivalent pruning results as under this comparison group – the input is the same, thus weight ranking only depends on weight magnitude. This finding further highlights the importance of using a proper comparison group for pruning LLMs, even for the classical magnitude pruning approach.

**Robustness to Calibration Samples.** We vary the number of calibration samples by selecting different sample sizes ranging between 1 and 256. Results are summarized in Figure 2. We see a clear difference in trend as the size of calibration data changes, where Wanda is much more robust when there are few calibration samples. Notably, even with a *single* sample, pruned networks obtained by Wanda have a perplexity of 7.66. This may be because input norm statistics $\|\mathbf{X}_j\|$ could be much easier to estimate than the full inverse hessian $\mathbf{H}^{-1}$ of the local layer-wise reconstruction problem.

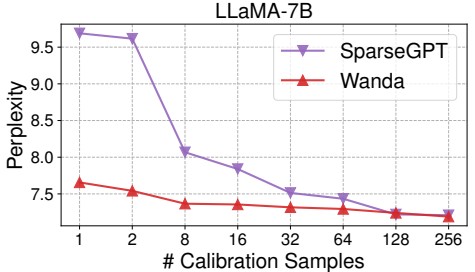

Figure 2: Wanda is more robust with less data.

**Weight Update.** We characterize the conditions under which the weight update process in SparseGPT can improve the effectiveness of pruning LLMs. We experiment with two ways of applying weight update: sequential and iterative. A sequential update means that at each layer, the full pruned mask is first computed and weight update is performed on the remaining weights. An iterative update means that the pruning and weight update steps proceed iteratively within each layer. SparseGPT adopts an iterative update procedure every 128 input channels, as it was found to give more accurate results.

Effects of the weight update on magnitude pruning and Wanda are summarized in Table 8. We study these two pruning methods because they do not involve any weight update by default. An iterative update changes the comparison group for unstructured pruning, which we denote in the table as (input, 128). We make several interesting observations:

- For all considered sparsities, weight update can improve magnitude pruning by a large margin.
- For unstructured 50% and 4:8 sparsities, weight update does not bring any improvement to Wanda.
- For 2:4 sparsity, the improvement (from 11.53 to 10.89) is marginal. Note that the best 2:4 sparse model (10.89) we obtained here is better than that obtained by SparseGPT (11.00 in Table 3).
- At 70% sparsity, weight update can bring notable improvement to Wanda (from 84.50 to 29.65), while the resulting pruned model (29.65) deteriorates a lot compared to dense LLaMA-7B (5.68).

| Pruning Configuration | | Weight Update | Sparsity | | | |
|---|---|---|---|---|---|---|
| Pruning Metric | Comparison Group | | 50% | 70% | 4:8 | 2:4 |
| Magnitude: $\|\mathbf{W}_{ij}\|$ | layer | ✗ | 17.59 | 5e4 | 16.84 | 42.13 |
| | layer | Sequential | **12.56** | 1e3 | **13.37** | **21.36** |
| | (input, 128) | Iterative | 26.77 | **3e2** | 36.98 | 47.61 |
| Wanda : $\|\mathbf{W}_{ij}\| \cdot \|\mathbf{X}_j\|$ | (output, 1) | ✗ | **7.26** | 84.50 | **8.57** | 11.53 |
| | (output, 1) | Sequential | 7.32 | 35.92 | 8.59 | **10.89** |
| | (input, 128) | Iterative | **7.26** | **29.65** | 8.68 | 11.43 |

Table 8: Effects of the weight update. It offers little or negligible improvement to Wanda.

## 6    RELATED WORK

**Network Pruning and Sparsity.** Pruning is a popular technique for compressing neural networks through the elimination of weights, yielding sparse networks (LeCun et al., 1989; Hassibi et al., 1993). It can be broadly categorized into *structured* and *unstructured* approaches.

Structured pruning methods (Liu et al., 2017; Molchanov et al., 2019; Fan et al., 2020; Shen et al., 2022; Xia et al., 2022; Fang et al., 2023; Nova et al., 2023), sometimes referred to as activation pruning (Gale et al., 2019; Dhillon et al., 2018), remove entire structured components of a network, facilitating efficient GPU speedups. Some existing methods (Babaeizadeh et al., 2016; Dubey et al., 2018) have explored structured pruning based on activation statistics of neuron/filter output, e.g. percentage of zero activations (Hu et al., 2016) and activation mean (Molchanov et al., 2017). Recently, Ma et al. (2023) have studied structured pruning of LLMs. Bansal et al. (2023); Liu et al. (2023b) and Elena Voita (2023) have demonstrated the existence of prompt-dependent and task-specific sparsity in the structural components of LLMs, e.g., attention heads and MLP neurons.

Unstructured methods (Han et al., 2015; 2016; Paul et al., 2023; Hoang et al., 2023; Gadhikar et al., 2023; Liu et al., 2023a) like magnitude pruning operate at the individual weight level, maintaining performance even at higher sparsity levels. Existing pruning methods usually require either modifications to the training procedure (Sanh et al., 2020; Kusupati et al., 2020), retraining the pruned networks to regain accuracy (Liu et al., 2019; Zhou et al., 2023), or an even more computationally intensive iterative retraining process (Renda et al., 2020; Frankle et al., 2020). However, scaling these methods to LLMs with billions of parameters presents a challenge, as the required training process demands substantial computational resources (Hoffmann et al., 2022; Zhang et al., 2022).

**Pruning with Limited Data.** Most related to our approach is a recent line of work on pruning with limited data (Hubara et al., 2021; Frantar et al., 2022; Frantar & Alistarh, 2022; Kwon et al., 2022). Such methods require no modification to the original training procedure and also no retraining of the pruned networks on the full training dataset. The primary aim of these methods is to preserve performance during the pruning procedure, assuming access to a limited and small amount of data, also referred to as the calibration data. In order to mitigate the accuracy drop, a layer-wise reconstruction problem (Hubara et al., 2021) is solved to minimize the change of output evaluated on the calibration data. Existing solvers (Singh & Alistarh, 2020; Frantar et al., 2022) for the layer-wise reconstruction problem rely on heavy computation of second-order Hessian inverses, which do not scale to the large hidden state size of LLMs. SparseGPT (Frantar & Alistarh, 2023) develops an efficient weight update procedure for LLMs via synchronized second-order Hessian updates.

**Emergent Properties of LLMs.** Our work is also related to recent studies on the existence of large magnitude outlier features in large language models (Kovaleva et al., 2021; Bondarenko et al., 2021; Timkey & Schijndel, 2021; Luo et al., 2021; Puccetti et al., 2022; Wei et al., 2022b). Dettmers et al. (2022) demonstrate that when LLMs exceed a certain parameter scale (e.g., 6B), large magnitude features start to emerge and strongly affect all layers, which can be seen as an emergent property of LLMs (Dettmers et al., 2022; Wei et al., 2022a; Schaeffer et al., 2023). They also pinpoint these emerging features as the reason why existing quantization methods fail. This observation has spurred the development of various quantization schemes (Dettmers et al., 2022; Xiao et al., 2023; Lin et al., 2023; Dettmers et al., 2023; Behdin et al., 2023) tailored specifically for LLMs to handle outlier features. Our work extends this understanding, demonstrating that outlier features should also serve as pivotal indicators of which weights to prune in LLMs.

## 7    CONCLUSION

In this work, we propose a simple and effective method for pruning Large Language Models (LLMs). Inspired by the recent discovery of emergent large magnitude features in LLMs, our approach, termed Wanda (Pruning by **W**eights **and a**ctivations), removes weights with the smallest magnitudes multiplied by the corresponding input activation norms, on a *per-output* basis. Without the need for any retraining or weight update procedures, Wanda is able to identify effective sparse networks within pretrained LLMs. We hope our work contributes to a better understanding of sparsity in LLMs. Last, considering the fast speed of pruning with Wanda, it would be interesting to investigate whether Wanda can be useful in the setting of sparse training (Evci et al., 2020; Peste et al., 2021; Kuznedelev et al., 2023; Benbaki et al., 2023; Frantar et al., 2023b), where pruning has to be conducted repeatedly and thus the pruning efficiency is critical.

**Acknowledgments.** We thank Yonghao Zhuang for valuable discussions. Mingjie Sun and Anna Bair were supported by funding from the Bosch Center for Artificial Intelligence.

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

## A    IMAGE CLASSIFIERS

We study how Wanda would perform against magnitude pruning on tasks where the latter has been widely used. We conduct a study on ImageNet-1K (Deng et al., 2009), a standard image classification task where magnitude pruning has been extensively studied (Gale et al., 2019; Blalock et al., 2020). We consider two modern vision architectures: ConvNeXt (Liu et al., 2022) and Vision Transformer (ViT) (Dosovitskiy et al., 2021). We choose these two architectures mainly for two reasons: first, as LLMs are based on Transformers, we would like to test if our observations on LLMs still hold on Transformers for other tasks; second, as we are evaluating on image classification, we are interested in examining how these pruning methods work on ConvNet models, with ConvNeXt being a representative architecture.

We use two ImageNet-1K pretrained models: ConvNeXt-B and DeiT-B, with a top-1 accuracy of 83.8% and 81.8% respectively. We prune the linear layers only (for ConvNeXt, this includes equivalent 1×1 convolution layers). For calibration data, we sample 4096 images from ImageNet training set. We observe that 4096 samples lead to a stable result for our pruning metric, beyond which we notice only a marginal effect. We report the accuracy of one-shot pruned models without any subsequent retraining.

We first study whether pruning *per output* is superior over pruning *per layer* for pruning image classifiers. In Figure 3, we show comparison results for both the magnitude metric and the pruning metric of Wanda. We can see that for both ConvNeXt-B and DeiT-B, layer-wise pruning is slightly better than pruning *per output*. We then compare the pruning metric of Wanda and the magnitude metric on layer-wise pruning. Results are shown in Figure 4. Our novel pruning metric leads to better results than magnitude pruning, especially at high sparsities (e.g., 70% and 80%).

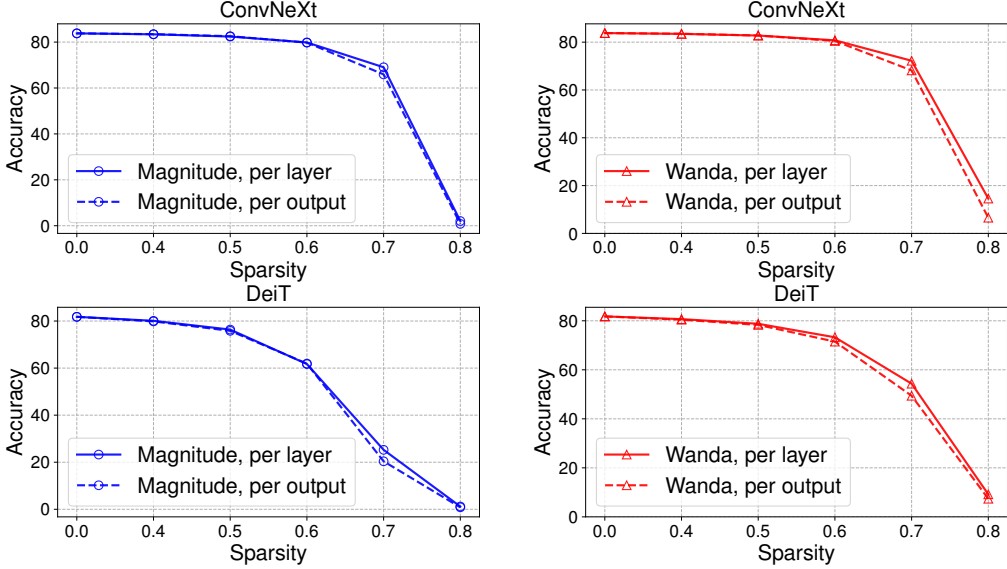

Figure 3: Analysis of comparison groups on pruning image classifiers.

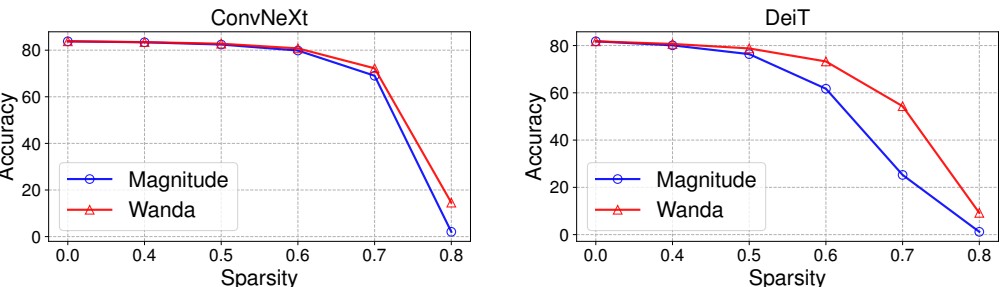

Figure 4: Our pruning metric outperforms the magnitude metric on pruning image classifiers.

## B WANDA ON PREVIOUS LLMS

In addition to LLaMA and LLaMA-2, we experiment with three previous LLM model families: namely OPT (Zhang et al., 2022), BLOOM (Scao et al., 2022) and Pythia (Biderman et al., 2023).

**Comparison with Baselines.** For OPT and Pythia, we experiment with varying sparsity levels (10% to 50%). We conduct additional evaluation on OPT and BLOOM models with various sizes. Results are shown in Table 9, Table 10 and Table 11 respectively. Our observations are as follows:

- Unlike LLaMA and LLaMA-2, the well-established magnitude pruning approach fails catastrophically on OPT-13B and Pythia-12B, even for low sparsity levels (e.g., 20%). This result further highlights the limitations of magnitude pruning for LLMs, as discussed in Section 3.

- Unlike magnitude pruning, Wanda successfully prunes these LLMs to much higher sparsities across various LLM model families, without any weight update on the kept weights. This result shows that LLMs have effective sub-networks that are *exact*. We hope this observation could contribute to a better understanding of sparsity in LLMs.

- There are cases where Wanda slightly underperforms SparseGPT, especially for OPT models (see Table 10), suggesting that for OPT, there may be a tradeoff between pruning speed and pruning accuracy. However, the gap between SparseGPT and Wanda tends to get smaller as model sizes increase. This can be seen in Table 10 and Table 11.

- At lower sparsities (e.g., 20%), Table 9 indicates that the computationally intensive weight update process may be unnecessary, as Wanda yields comparable or slightly superior results.

| Model | Dense | Pruning Method | Weight Update | Sparsity | | | | |
|---|---|---|---|---|---|---|---|---|
| | | | | 10% | 20% | 30% | 40% | 50% |
| | | Magnitude | ✗ | 14.45 | 9e3 | 1e4 | 1e4 | 1e4 |
| OPT-13B | 10.13 | SparseGPT | ✓ | 10.11 | 10.10 | 10.12 | **10.35** | **11.19** |
| | | Wanda | ✗ | **10.09** | **10.07** | **10.09** | 10.63 | 11.42 |
| | | Magnitude | ✗ | 127.76 | 2e5 | 7e5 | 2e5 | 3e5 |
| Pythia-12B | 8.59 | SparseGPT | ✓ | **8.59** | 8.65 | 8.86 | 9.39 | **11.02** |
| | | Wanda | ✗ | **8.59** | **8.60** | **8.85** | **9.31** | 11.27 |

Table 9: Pruning Pythia-13B and OPT-13B with various sparsity levels.

| Method | Weight Update | Sparsity | OPT | | | | | |
|---|---|---|---|---|---|---|---|---|
| | | | 125m | 350m | 1.3B | 2.7B | 6.7B | 13B |
| Dense | - | 50% | 27.66 | 22.00 | 14.62 | 12.47 | 10.86 | 10.13 |
| Magnitude | ✗ | 50% | 7e3 | 6e3 | 1e4 | 9e3 | 9e4 | 2e4 |
| SparseGPT | ✓ | 50% | **37.07** | **34.76** | **17.44** | **13.48** | **11.57** | **11.19** |
| Wanda | ✗ | 50% | 38.96 | 35.92 | 19.12 | 14.28 | 11.94 | 11.42 |

Table 10: Pruning OPT family models with various sizes.

| Method | Weight Update | Sparsity | BLOOM | | | | |
|---|---|---|---|---|---|---|---|
| | | | 560m | 1.1B | 1.7B | 3B | 7.1B |
| Dense | - | 50% | 22.42 | 17.68 | 15.39 | 13.48 | 11.37 |
| Magnitude | ✗ | 50% | 2e10 | 1e6 | 2e5 | 8e6 | 2e6 |
| SparseGPT | ✓ | 50% | **28.92** | **21.35** | **18.88** | 16.76 | 13.96 |
| Wanda | ✗ | 50% | 30.74 | 22.72 | 19.79 | **16.45** | **13.55** |

Table 11: Pruning BLOOM family models with various sizes.

**Comparison Group** We test if our observation regarding pruning *per output* holds true for other LLM model families. We experiment on OPT (Zhang et al., 2022) and BLOOM (Scao et al., 2022). In Table 12 and Table 13, we provide results comparing pruning *per layer* and pruning *per output* for these two LLM model families. The pruning metric is fixed to be our proposed metric: $|\mathbf{W}_{ij}| \cdot \|\mathbf{X}_j\|$. We can see that our findings regarding the comparison group are not limited to LLaMA. For OPT and BLOOM model families, pruning *per output* consistently outperforms pruning *per layer*.

| Comparison Group | Sparsity | OPT | | | | | |
|---|---|---|---|---|---|---|---|
| | | 125m | 350m | 1.3B | 2.7B | 6.7B | 13B |
| *per layer* | 50% | 46.95 | 38.97 | 22.20 | 22.66 | 15.35 | 13.54 |
| *per output* | 50% | **38.96** | **36.19** | **19.42** | **14.22** | **11.97** | **11.42** |

Table 12: Comparison of pruning *per layer* versus *per output* for OPT models.

| Comparison Group | Sparsity | BLOOM | | | | |
|---|---|---|---|---|---|---|
| | | 560m | 1.1B | 1.7B | 3B | 7.1B |
| *per layer* | 50% | 34.57 | 26.26 | 22.55 | 18.22 | 15.31 |
| *per output* | 50% | **30.74** | **22.72** | **19.79** | **16.45** | **13.55** |

Table 13: Comparison of pruning *per layer* versus *per output* for BLOOM models.

## C  ADDITIONAL BASELINES

We compare with several prior activation pruning methods. These approaches remove entire neurons in the network based on certain statistics of the neuron output: mean and standard deviation (Molchanov et al., 2017), correlation (Babaeizadeh et al., 2016) and mean squared norm (Dubey et al., 2018). We show the results of pruning LLaMA-7B in Table 14. We compute these output statistics using the calibration set and remove neurons with smaller values. We observe that these activation pruning methods are unable to prune LLMs effectively.

We also compare with several prior methods on pruning BERT (Devlin et al., 2018). In Table 15, we provide a summary of existing pruning methods, mostly for pruning BERT. A key distinction of these methods and our work is that they interleave pruning heavily with the fine-tuning process. Another difference is that BERT pruning methods focus on performance on a downstream task, rather than preserving the general performance of pretrained language models.

We adopt these prior methods for pruning LLMs, where the goal is to preserve the language modeling ability. Thus we use the pre-training auto-regressive loss to compute their pruning metrics. We evaluate two settings: one-shot pruning and one-shot pruning followed by fine-tuning. For one-shot pruning, we use the pruning metrics listed in Table 15 to prune LLMs. We fine-tune the pruned LLMs within a limited computational budget, i.e., one day. Results are summarized in Table 16. We observe that these pruning methods are not effective when adapted for pruning LLMs.

| Model | Dense | Activation Statistics | Sparsity | | | | |
|---|---|---|---|---|---|---|---|
| | | | 10% | 20% | 30% | 40% | 50% |
| LLaMA-7B | 5.68 | Mean | 1e5 | 2e5 | 3e5 | 3e5 | 3e5 |
| | | Standard Deviation | 161 | 649 | 7e3 | 1e5 | 2e5 |
| | | Correlation | 1e4 | 7e4 | 2e5 | 2e5 | 2e5 |
| | | Mean Squared Norm | 16.43 | 98.13 | 9e2 | 1e5 | 4e5 |

Table 14: Results for activation pruning methods.

| Pruning Method | Pruning Type | Pruning Metric | Training Procedure |
|---|---|---|---|
| SNIP (Lee et al., 2018) | Unstructured | Loss Sensitivity | Pruning at Initialization |
| BERT-LTH (Chen et al., 2020) | Unstructured | Magnitude | Fine-tuning BERT |
| Movement (Sanh et al., 2020) | Unstructured | Loss Sensitivity | Fine-tuning BERT |
| Platon (Zhang et al., 2021) | Unstructured | Loss Sensitivity | Fine-tuning BERT |
| PINS (Ren & Zhu, 2023) | Unstructured | Loss Sensitivity | Fine-tuning BERT |

Table 15: Summary of prior pruning methods on BERT.

| Model | Dense | Fine-tuning | Pruning method | | | | | |
|---|---|---|---|---|---|---|---|---|
| | | | SNIP | BERT-LTH | Movement | Platon | PINS | Wanda |
| LLaMA-7B | 5.68 | ✗ | 231.48 | 17.29 | 349.33 | 124.91 | 89.12 | **7.26** |
| | | ✓ | 102.32 | 12.43 | 168.17 | 102.34 | 72.13 | **6.28** |

Table 16: Comparisons with prior pruning methods on BERT (unstructured 50% sparsity).

# D  COMPLEMENTARY EXPERIMENTAL RESULTS

In this section, we supplement the main paper with additional experimental results. This includes analysis on the number of calibration samples (Appendix D.1), robustness analysis under random seeds (Appendix D.2), evaluation at higher unstructured sparsity levels (Appendix D.3), few-shot results (Appendix D.4) and a detailed performance breakdown for zero-shot tasks (Appendix D.5 and Appendix D.6).

## D.1  NUMBER OF CALIBRATION SAMPLES

In the main paper, the default number of calibration samples is 128. This choice is adopted from Frantar & Alistarh (2023), which was selected on the OPT model family (Zhang et al., 2022). Here we conduct a detailed analysis on the effect of the number of calibration samples for LLaMA and LLaMA-2 model families. We show the results for pruning LLaMA-7B and LLaMA-2-7B with unstructured 50% sparsity in Table 17. We find that there is a slight improvement in performance of pruned LLMs when the size of calibration set goes beyond 128.

| Model | Method | 1 | 16 | 32 | 64 | 128 | 256 | 512 | 1024 | 2048 |
|---|---|---|---|---|---|---|---|---|---|---|
| LLaMA-7B | SparseGPT | 10.22 | 7.61 | 7.36 | 7.29 | 7.26 | 7.20 | 7.19 | 7.23 | 7.20 |
| | Wanda | 7.46 | 7.27 | 7.28 | 7.28 | 7.26 | 7.30 | 7.26 | 7.25 | 7.26 |
| LLaMA-2-7B | SparseGPT | 8.63 | 6.67 | 6.62 | 6.61 | 6.53 | 6.52 | 6.50 | 6.49 | 6.49 |
| | Wanda | 6.53 | 6.45 | 6.46 | 6.45 | 6.45 | 6.45 | 6.45 | 6.45 | 6.45 |

Table 17: WikiText validation perplexity of pruned LLaMA and LLaMA-2 under various number of calibration samples, with 50% sparsity.

## D.2  ROBUSTNESS ANALYSIS

In this part, we perform a robustness analysis of our results in Section 4.2. The result in Table 3 is evaluated under a fixed calibration set. Since both SparseGPT and Wanda require calibration data to estimate input statistics, we sample different calibration sets under 5 random seeds and evaluate these two pruning methods. In Table 18, we report the perplexity (mean and standard deviation) of pruned LLaMA models under 5 random seeds. In many cases, the variance across random seeds is lower for Wanda, suggesting that Wanda is more stable with variations in the calibration sets.

| Method | Weight Update | Sparsity | LLaMA 7B | LLaMA 13B | LLaMA-2 7B | LLaMA-2 13B |
|---|---|---|---|---|---|---|
| Dense | - | 0% | 5.68 | 5.09 | 5.12 | 4.57 |
| Magnitude | ✗ | 50% | 17.29 | 20.21 | 14.89 | 6.37 |
| SparseGPT | ✓ | 50% | **7.25** (±0.03) | 6.24 (±0.02) | 6.52 (±0.02) | 5.63 (±0.01) |
| Wanda | ✗ | 50% | **7.25** (±0.01) | **6.18** (±0.01) | **6.44** (±0.01) | **5.59** (±0.01) |
| Magnitude | ✗ | 4:8 | 16.84 | 13.84 | 16.48 | 6.76 |
| SparseGPT | ✓ | 4:8 | 8.67 (±0.08) | **7.43** (±0.03) | 8.05 (±0.03) | 6.59 (±0.04) |
| Wanda | ✗ | 4:8 | **8.65** (±0.01) | **7.43** (±0.03) | **7.98** (±0.01) | **6.56** (±0.01) |
| Magnitude | ✗ | 2:4 | 42.13 | 18.37 | 54.59 | 8.33 |
| SparseGPT | ✓ | 2:4 | **10.94** (±0.23) | **9.08** (±0.04) | **10.44** (±0.42) | **8.28** (±0.05) |
| Wanda | ✗ | 2:4 | 11.48 (±0.05) | 9.60 (±0.04) | 11.10 (±0.09) | **8.28** (±0.02) |

Table 18: WikiText validation perplexity of pruned LLaMA and LLaMA-2 models. We report the mean and standard deviation under 5 random seeds.

## D.3 HIGHER SPARSITY

In Section 4, we have evaluated unstructured pruning with a sparsity level of 50%. This is to follow the evaluation setup of Frantar & Alistarh (2023). In this part, we evaluate on higher sparsity levels, i.e., 60% and 80%. Results for these two sparsity levels are shown Table 19 and Table 20 respectively. At 60% sparsity, Wanda remains competitive with SparseGPT. At 80% sparsity, SparseGPT is able to outperform Wanda, but the performance drop compared to the dense counterpart is significant. The best 80% sparse model (25.86) underperforms the smallest dense LLaMA-7B model (5.68) by a large gap. This suggests that at extreme sparsity levels, it may be better to use a small dense model trained to convergence instead.

| Method | Weight Update | Sparsity | LLaMA 7B | LLaMA 13B | LLaMA 30B | LLaMA 65B | LLaMA-2 7B | LLaMA-2 13B | LLaMA-2 70B |
|---|---|---|---|---|---|---|---|---|---|
| Dense | - | 0% | 5.68 | 5.09 | 4.77 | 3.56 | 5.12 | 4.57 | 3.12 |
| Magnitude | ✗ | 60% | 6e2 | 2e2 | 27.67 | 9.34 | 4e3 | 11.23 | 8.21 |
| SparseGPT | ✓ | 60% | **10.51** | **8.56** | 6.66 | **5.82** | **9.58** | 7.80 | **4.98** |
| Wanda | ✗ | 60% | 10.66 | **8.56** | **6.49** | 5.83 | 9.71 | **7.75** | **4.98** |

Table 19: WikiText validation perplexity of pruned LLaMA and LLaMA-2 models with unstructured 60% sparsity.

| Method | Weight Update | Sparsity | LLaMA 7B | LLaMA 13B | LLaMA 30B | LLaMA 65B | LLaMA-2 7B | LLaMA-2 13B | LLaMA-2 70B |
|---|---|---|---|---|---|---|---|---|---|
| Dense | - | 0% | 5.68 | 5.09 | 4.77 | 3.56 | 5.12 | 4.57 | 3.12 |
| Magnitude | ✗ | 80% | 1e5 | 3e4 | 1e5 | 2e4 | nan | 5e4 | 3e4 |
| SparseGPT | ✓ | 80% | **2e2** | **1e2** | **54.98** | **32.80** | **1e2** | **1e2** | **25.86** |
| Wanda | ✗ | 80% | 5e3 | 4e3 | 2e3 | 2e3 | 5e3 | 2e3 | 1e2 |

Table 20: WikiText validation perplexity of pruned LLaMA and LLaMA-2 models with unstructured 80% sparsity.

## D.4 FEW-SHOT RESULTS ON MMLU

Our experiments in Section 4.1 focus on zero-shot evaluation. However, LLMs are also known for their ability to learn in context. In this part, we conduct additional evaluation on few-shot tasks. Specifically,

we choose the Massive Multitask Language Understanding benchmark (MMLU) (Hendrycks et al., 2021). In alignment with the evaluation methodology of Touvron et al. (2023a), we perform 5-shot evaluation. In Table 21, we report the mean accuracies for both dense LLMs and sparse LLMs with unstructured 50% sparsity. In the few-shot setting, Wanda performs competitively with SparseGPT. Notably, large sparse LLMs surpass smaller dense counterparts, e.g., sparse LLaMA-13B/LLaMA-2-13B versus dense LLaMA-7B/LLaMA-2-7B. This trend can not be observed from the standard magnitude pruning approach.

| | | | LLaMA | | LLaMA-2 | |
|---|---|---|---|---|---|---|
| Method | Weight Update | Sparsity | 7B | 13B | 7B | 13B |
| Dense | - | 0% | 39.85 | 52.92 | 52.08 | 61.69 |
| Magnitude | ✗ | 50% | 30.69 | 30.69 | 32.14 | 48.76 |
| SparseGPT | ✓ | 50% | **34.43** | 45.08 | 38.68 | 54.83 |
| Wanda | ✗ | 50% | 33.49 | **46.04** | **39.27** | **55.01** |

Table 21: 5-shot results (mean accuracies %) on MMLU for unstructured 50% sparsity.

## D.5 FINE-TUNING

In Table 6 of Section 5, we report the mean zero-shot accuracies after fine-tuning Wanda pruned LLaMA-7B models. In this part, we report the task-wise performance of these fine-tuned models. Results are summarized in Table 22. For per-task accuracies, most of the performance drop during pruning can be recovered through fine-tuning. Note that here we are performing limited fine-tuning with a computational budget (12 hours for LoRA fine-tuning and 3 days for full parameter fine-tuning). It remains to be seen if the gap between sparse pruned LLMs and the dense counterparts can be fully recovered given more computational budget.

| Sparsity | Fine-tuning | BoolQ | RTE | HellaSwag | WinoGrande | ARC-e | ARC-c | OBQA | Mean |
|---|---|---|---|---|---|---|---|---|---|
| Dense | - | 75.05 | 66.43 | 56.92 | 69.93 | 75.34 | 41.89 | 34.40 | 59.99 |
| 50% | ✗ | 71.22 | 55.60 | 51.85 | 66.06 | 69.11 | 36.86 | 28.80 | 54.21 |
| | LoRA | 72.90 | 60.79 | 55.36 | 67.48 | 71.42 | 37.97 | 29.80 | 56.53 |
| | Full | **74.50** | **62.84** | **55.83** | **69.02** | **73.49** | **39.20** | **32.20** | **58.15** |
| 4:8 | ✗ | 70.97 | 58.24 | 46.81 | 65.83 | 65.53 | 33.97 | 28.00 | 52.76 |
| | LoRA | 71.24 | 60.04 | 54.47 | 66.14 | 67.68 | 35.32 | 29.20 | 54.87 |
| | Full | **73.32** | **60.99** | **55.21** | **66.80** | **71.76** | **36.46** | **32.00** | **56.65** |
| 2:4 | ✗ | 69.30 | 51.99 | 42.06 | 62.75 | 60.94 | 28.07 | 24.60 | 48.53 |
| | LoRA | 70.32 | **64.98** | 52.53 | 65.04 | 67.00 | 33.53 | 27.80 | 54.46 |
| | Full | **73.21** | 61.34 | **54.86** | **66.18** | **70.24** | **35.68** | **31.80** | **56.19** |

Table 22: The gap between pruned LLMs and dense LLMs can be largely mitigated via fine-tuning.

## D.6 ZERO-SHOT TASKS

For zero-shot results in Section 4.1, the 7 evaluated zero-shot tasks are: BoolQ (Clark et al., 2019), RTE (Wang et al., 2018), HellaSwag (Zellers et al., 2019), WinoGrande (Sakaguchi et al., 2019), ARC Easy and Challenge (Clark et al., 2018), and OpenbookQA (Mihaylov et al., 2018). For reproducibility, we used commit `df3da98` on the main branch. All tasks were evaluated on task version 0 except for BoolQ, where the evaluated version was 1. We show the task-wise performance in Table 23,24,25,26,27 and 28.

| Params | Method | BoolQ | RTE | HellaSwag | WinoGrande | ARC-e | ARC-c | OBQA | Mean |
|---|---|---|---|---|---|---|---|---|---|
| | Dense | 75.05 | 66.43 | 56.92 | 69.93 | 75.34 | 41.89 | 34.40 | 59.99 |
| 7B | Magnitude | 54.59 | 54.51 | 45.49 | 59.19 | 58.84 | 33.53 | 22.40 | 46.94 |
| | SparseGPT | **72.05** | 54.15 | 51.43 | **67.88** | **71.38** | **37.71** | **30.00** | **54.94** |
| | Wanda | 71.22 | **55.60** | **51.85** | 66.06 | 69.11 | 36.86 | 28.80 | 54.21 |
| | Dense | 77.89 | 70.4 | 59.94 | 72.77 | 77.40 | 46.50 | 33.20 | 62.59 |
| 13B | Magnitude | 54.89 | 51.26 | 44.16 | 63.14 | 58.80 | 33.79 | 27.20 | 47.61 |
| | SparseGPT | **76.97** | 61.01 | 54.95 | 71.67 | 72.47 | 41.98 | 31.20 | 58.61 |
| | Wanda | 75.90 | **62.82** | **55.71** | **71.98** | **73.19** | **43.52** | **32.20** | **59.33** |
| | Dense | 82.69 | 66.79 | 63.35 | 75.69 | 80.30 | 52.82 | 36.00 | 65.38 |
| 30B | Magnitude | 64.34 | 50.18 | 50.59 | 66.54 | 72.39 | 43.77 | 29.00 | 53.83 |
| | SparseGPT | **82.32** | 62.45 | 59.15 | **75.22** | 78.96 | 48.56 | **35.00** | 63.09 |
| | Wanda | 81.90 | **65.34** | **60.93** | 73.48 | **79.29** | **49.66** | 34.60 | **63.60** |
| | Dense | 84.83 | 69.68 | 64.54 | 77.27 | 81.40 | 52.90 | 38.20 | 66.97 |
| 65B | Magnitude | 79.15 | 62.45 | 61.90 | 74.74 | 76.40 | 49.57 | 35.00 | 62.74 |
| | SparseGPT | 84.60 | 70.76 | 63.90 | **77.43** | 79.35 | **50.85** | 37.20 | 66.30 |
| | Wanda | **84.70** | **71.48** | **64.55** | 76.87 | **79.75** | 50.51 | **38.80** | **66.67** |

Table 23: Accuracies (%) of LLaMA for 7 zero-shot tasks with unstructured 50% sparsity.

| Params | Method | BoolQ | RTE | HellaSwag | WinoGrande | ARC-e | ARC-c | OBQA | Mean |
|---|---|---|---|---|---|---|---|---|---|
| | Dense | 75.05 | 66.43 | 56.92 | 69.93 | 75.34 | 41.89 | 34.40 | 59.99 |
| 7B | Magnitude | 51.19 | 50.54 | 46.73 | 60.69 | 58.96 | 30.89 | 23.20 | 46.03 |
| | SparseGPT | **73.06** | 58.12 | **47.88** | **65.98** | **66.75** | 32.42 | 25.40 | **52.80** |
| | Wanda | 70.97 | **58.24** | 46.81 | 65.83 | 65.53 | **33.97** | **28.00** | 52.76 |
| | Dense | 77.89 | 70.40 | 59.94 | 72.77 | 77.40 | 46.50 | 33.20 | 62.59 |
| 13B | Magnitude | 61.07 | 51.26 | 48.91 | 65.11 | 63.26 | 35.67 | 28.40 | 50.53 |
| | SparseGPT | **76.61** | 57.76 | 51.24 | 70.17 | **71.17** | 37.20 | 27.80 | 55.99 |
| | Wanda | 74.89 | **57.89** | **51.26** | **70.56** | 70.29 | **37.97** | **29.80** | **56.09** |
| | Dense | 82.69 | 66.79 | 63.35 | 75.69 | 80.30 | 52.82 | 36.00 | 65.38 |
| 30B | Magnitude | 63.55 | 50.18 | 49.45 | 65.75 | 73.36 | 42.83 | 29.60 | 53.53 |
| | SparseGPT | **78.69** | **61.73** | 56.15 | **74.35** | 76.94 | 46.08 | 31.60 | 60.79 |
| | Wanda | 77.38 | 58.80 | **58.79** | 74.28 | **77.34** | **46.46** | **34.00** | **61.00** |
| | Dense | 84.83 | 69.68 | 64.54 | 77.27 | 81.40 | 52.90 | 38.20 | 66.97 |
| 65B | Magnitude | 74.95 | 68.23 | 60.85 | 74.27 | 76.45 | 47.61 | 32.80 | 62.17 |
| | SparseGPT | **84.35** | 68.95 | **61.00** | **77.19** | 78.75 | 48.46 | 35.40 | 64.87 |
| | Wanda | 84.29 | **70.92** | 59.54 | 76.64 | **79.00** | **48.83** | **35.60** | **64.97** |

Table 24: Accuracies (%) of LLaMA for 7 zero-shot tasks with 4:8 sparsity.

| Params | Method | BoolQ | RTE | HellaSwag | WinoGrande | ARC-e | ARC-c | OBQA | Mean |
|---|---|---|---|---|---|---|---|---|---|
| | Dense | 75.05 | 66.43 | 56.92 | 69.93 | 75.34 | 41.89 | 34.40 | 59.99 |
| 7B | Magnitude | 53.09 | 55.60 | 42.30 | 59.91 | 53.28 | 27.13 | 21.80 | 44.73 |
| | SparseGPT | **70.46** | **60.65** | **42.99** | **64.88** | **61.49** | **30.12** | 23.60 | **50.60** |
| | Wanda | 69.30 | 51.99 | 42.06 | 62.75 | 60.94 | 28.07 | **24.60** | 48.53 |
| | Dense | 77.89 | 70.40 | 59.94 | 72.77 | 77.40 | 46.50 | 33.20 | 62.59 |
| 13B | Magnitude | 60.95 | 49.10 | 45.81 | 62.75 | 58.75 | 31.06 | 27.60 | 48.00 |
| | SparseGPT | **72.14** | **55.23** | **48.11** | **68.98** | **66.71** | **34.98** | 26.40 | **53.22** |
| | Wanda | 70.21 | 53.43 | 46.74 | 68.82 | 65.82 | 33.87 | **27.20** | 52.30 |
| | Dense | 82.69 | 66.79 | 63.35 | 75.69 | 80.30 | 52.82 | 36.00 | 65.38 |
| 30B | Magnitude | 65.11 | 52.35 | 51.72 | 66.22 | 70.88 | 38.23 | 27.60 | 53.16 |
| | SparseGPT | **75.60** | 62.13 | 53.10 | 72.61 | **75.13** | 41.98 | 31.80 | 58.91 |
| | Wanda | 74.68 | **63.80** | **54.41** | **72.93** | 74.41 | **42.06** | **32.20** | **59.21** |
| | Dense | 84.83 | 69.68 | 64.54 | 77.27 | 81.40 | 52.90 | 38.20 | 66.97 |
| 65B | Magnitude | 77.9 | 64.98 | 58.65 | 72.85 | 75.15 | 45.05 | 34.40 | 61.28 |
| | SparseGPT | 83.15 | 65.34 | **57.20** | **76.72** | 78.20 | 45.18 | 32.20 | 62.57 |
| | Wanda | **83.58** | **66.79** | 56.36 | 75.82 | **78.23** | **45.56** | **33.60** | **62.84** |

Table 25: Accuracies (%) of LLaMA for 7 zero-shot tasks with 2:4 sparsity.

| Params | Method | BoolQ | RTE | HellaSwag | WinoGrande | ARC-e | ARC-c | OBQA | Mean |
|---|---|---|---|---|---|---|---|---|---|
| | Dense | 77.74 | 62.82 | 57.17 | 68.90 | 76.39 | 43.52 | 31.40 | 59.71 |
| 7B | Magnitude | 63.00 | **57.04** | 49.13 | 63.30 | 64.10 | 34.64 | 26.80 | 51.14 |
| | SparseGPT | 75.02 | 54.15 | 52.37 | **69.85** | **73.27** | **39.85** | 29.20 | **56.24** |
| | Wanda | **75.99** | 53.43 | **52.49** | 68.19 | 72.77 | 39.59 | **31.20** | **56.24** |
| | Dense | 80.52 | 65.34 | 60.06 | 72.22 | 79.42 | 48.46 | 35.20 | 63.03 |
| 13B | Magnitude | 57.61 | 55.96 | 54.40 | 65.27 | 70.54 | 38.40 | 27.80 | 52.85 |
| | SparseGPT | 81.44 | **65.34** | 55.83 | **72.77** | 74.83 | 42.24 | 32.60 | 60.72 |
| | Wanda | **81.84** | 64.02 | **56.90** | 71.35 | **76.18** | **43.52** | 32.00 | **60.83** |
| | Dense | 83.40 | 67.87 | 66.10 | 78.06 | 82.55 | 54.44 | 37.20 | 67.08 |
| 70B | Magnitude | 70.55 | 60.65 | 61.50 | 73.48 | 75.70 | 49.23 | 35.40 | 60.93 |
| | SparseGPT | **83.55** | 70.40 | 63.80 | **78.85** | **82.40** | **53.75** | **38.20** | **67.28** |
| | Wanda | 82.50 | **73.65** | **64.10** | 78.14 | 80.80 | 52.65 | 37.40 | 67.03 |

Table 26: Accuracies (%) of LLaMA-2 for 7 zero-shot tasks with unstructured 50% sparsity.

| Params | Method | BoolQ | RTE | HellaSwag | WinoGrande | ARC-e | ARC-c | OBQA | Mean |
|---|---|---|---|---|---|---|---|---|---|
| | Dense | 77.74 | 62.82 | 57.17 | 68.90 | 76.39 | 43.52 | 31.40 | 59.71 |
| 7B | Magnitude | 63.00 | 52.35 | 50.08 | 62.43 | 64.73 | 35.92 | 26.00 | 50.64 |
| | SparseGPT | 72.69 | **55.23** | **48.20** | **68.11** | **69.15** | **35.84** | **27.40** | **53.80** |
| | Wanda | **73.91** | 53.79 | 46.45 | 66.61 | 66.71 | 34.13 | 25.80 | 52.49 |
| | Dense | 80.52 | 65.34 | 60.06 | 72.22 | 79.42 | 48.46 | 35.20 | 63.03 |
| 13B | Magnitude | 63.33 | 57.76 | 53.96 | 64.40 | 68.48 | 35.75 | 26.00 | 52.81 |
| | SparseGPT | 79.97 | **66.79** | 52.01 | **70.64** | 73.61 | 41.04 | **30.00** | **59.15** |
| | Wanda | **80.26** | 65.62 | **52.05** | 69.48 | **73.88** | **41.54** | 28.40 | 58.75 |
| | Dense | 83.40 | 67.87 | 66.10 | 78.06 | 82.55 | 54.44 | 37.20 | 67.08 |
| 70B | Magnitude | 70.95 | 59.21 | 60.05 | 74.11 | 76.25 | 46.76 | 34.60 | 60.28 |
| | SparseGPT | 82.20 | **72.20** | 61.45 | **77.82** | **80.85** | 51.19 | 35.20 | 65.84 |
| | Wanda | **84.30** | 71.80 | **61.90** | 76.24 | 80.40 | **51.80** | **36.00** | **66.06** |

Table 27: Accuracies (%) of LLaMA-2 for 7 zero-shot tasks with 4:8 sparsity.

| Params | Method | BoolQ | RTE | HellaSwag | WinoGrande | ARC-e | ARC-c | OBQA | Mean |
|--------|--------|-------|-----|-----------|------------|-------|-------|------|------|
|        | Dense  | 77.74 | 62.82 | 57.17 | 68.90 | 76.39 | 43.52 | 31.40 | 59.71 |
| 7B     | Magnitude | 56.23 | 51.35 | 42.27 | 60.93 | 59.18 | 27.31 | 21.80 | 45.58 |
|        | SparseGPT | **70.52** | **58.84** | **43.26** | **66.69** | **64.10** | 29.97 | 23.20 | **50.94** |
|        | Wanda | 67.65 | 53.07 | 40.92 | 62.43 | 61.78 | **31.20** | **24.20** | 48.75 |
|        | Dense  | 80.52 | 65.34 | 60.06 | 72.22 | 79.42 | 48.46 | 35.20 | 63.03 |
| 13B    | Magnitude | 65.69 | 54.15 | 50.13 | 62.04 | 62.46 | 31.74 | 23.00 | 49.89 |
|        | SparseGPT | 76.79 | 59.38 | 46.58 | **68.67** | **70.62** | 36.60 | 25.40 | 54.86 |
|        | Wanda | **76.80** | **61.22** | **47.82** | 66.90 | 69.24 | **36.82** | **26.40** | **55.03** |
|        | Dense  | 83.40 | 67.87 | 66.10 | 78.06 | 82.55 | 54.44 | 37.20 | 67.08 |
| 70B    | Magnitude | 73.20 | 57.04 | 58.40 | 74.27 | 76.15 | 45.22 | 35.40 | 59.95 |
|        | SparseGPT | 79.50 | **70.76** | 59.00 | **76.64** | 78.95 | **48.55** | 33.80 | 63.89 |
|        | Wanda | **82.20** | 69.85 | **59.34** | 76.23 | **79.30** | 47.26 | **34.80** | **64.14** |

Table 28: Accuracies (%) of LLaMA-2 for 7 zero-shot tasks with 2:4 sparsity.

