# OpenReview forum: "A Simple and Effective Pruning Approach for Large Language Models"
_ICLR.cc/2024/Conference — ICLR 2024 poster_

### Official Review · Reviewer_V88x · 2023-10-27

**Soundness:** 3 good
**Presentation:** 3 good
**Contribution:** 2 fair
**Rating:** 6
**Confidence:** 4

**Summary:**

This study introduces Wanda, an unstructured pruning method for large language models (LLMs). Wanda utilizes both weights and activations as proxies to estimate the importance of parameters, selectively zeroing out those with low importance. This approach results in a sparse LLM, which can be accelerated with some special devices capable of 2:4 sparsity acceleration. Wanda not only outperforms traditional magnitude-based pruning methods for efficiency but also offers quicker processing compared to SparseGPT, which depends on second-order information.

**Strengths:**

- The method proposed is simple and impressively effective, particularly when compared to SparseGPT. Its lower computational complexity is a significant benefit, leading to reduced memory requirements during the pruning process.
- This paper is well-organized and easy to follow. The authors thoughtfully include extensive experimental results from LLMs such as LLaMA, LLaMA-2, OPT, and Bloom, providing a useful resource for future research in this field.

**Weaknesses:**

- My primary concern lies in the relative contribution of Wanda compared to SparseGPT and several existing methods. There are a lot of criteria in the literature of pruning, such as first-order Taylor expansion ($|w*\delta w|$) [1], and optimal brain damage [2]. It seems that these general methods, with a similar form to Wanda, can be easily adapted to LLM. So, is the proposed $|w| \cdot |x|$ the optimal criterion, especially in the context of LLMs?

- It would be helpful if the authors could detail the overall speed-up of the entire LLM. Table 6 indeed summarizes the speedup for a single layer. However, an LLM usually comprises various components, including QKV, attention, and linear layers. Since Wanda primarily targets the sparsification of linear layers, it may not significantly boost attention computation speeds. Therefore, the actual overall speedup could differ from the single-layer improvements indicated in Table 6.

- The effectiveness of Wanda on models like OPT and Pythia, particularly under high sparsity conditions (e.g., 50%), seems to be limited. In these scenarios, SparseGPT has even demonstrated superior performance.

[1] Molchanov, Pavlo, et al. "Importance estimation for neural network pruning." Proceedings of the IEEE/CVF conference on computer vision and pattern recognition. 2019.
[2] LeCun, Yann, John Denker, and Sara Solla. "Optimal brain damage." Advances in neural information processing systems 2 (1989).

**Questions:**

My questions can be found in the weaknesses above.

---

> ### Author Response · Authors · 2023-11-17
> **Response**
>
> We thank the reviewer for the positive assessment of our paper and the constructive comments. We are happy to address your concerns.
> - **Relative contribution compared to Taylor Pruning [1]**
> In our submitted version, we have cited [1] in the related work as a structured pruning method. To obtain the pruning metric on a convolutional filter, [1] uses the first-order or second-order Taylor expansion approximation of the loss function. The metric $|\delta w| \cdot |w|$ for a convolutional filter corresponds to the first-order Taylor expansion.
>
>   The major difference between [1] and our work is that [1] is focused on filter-level pruning of convolutional neural networks. Wanda is a weight-level pruning method. Due to the difference in the pruning granularity (i.e., the set of weights that are removed together), the pruned networks can have different properties, e.g. sparsity level and inference speedup. Thus in existing pruning literature, these two types of pruning methods have been separately investigated and compared against within each branch.
>
>   Last, the use of the pruning metric $|\delta w| \cdot |w|$ for weight-level pruning has been proposed by SNIP [3]. Specifically, SNIP uses this criterion to prune neural networks at initializations and show that the pruned networks obtained by $|\delta w| \cdot |w|$ can be trained from scratch to high accuracy. In our paper, we have compared against this pruning metric in the Appendix (section *Additional Baselines*). We find that this metric, while initially proposed in [3] to find sparse trainable networks, is not effective when pruning LLMs.
>
> - **Relative contribution compared to SparseGPT and Optimal Brain Damage [2]**
> Our method is more related to weight-level pruning methods SparseGPT and Optimal Brain Damage (OBD) [2]. In the *Remark* paragraph of section 3, we have discussed in detail the relationship between Wanda, SparseGPT and OBD. We show that how the pruning metrics of Wanda and SparseGPT can be related with reduction steps similar to the idea of OBD, i.e. dropping the off-diagonal elements of the Hessian matrix.
>
>   The difference to OBD can be attributed to the fact that OBD adopts the pruning metric that considers the global training objective. This is the main reason why OBD is impractical for modern neural networks. For the Hessian of the global training loss function on network parameters, computing the diagonal elements of a Hessian is fundamentally just as expensive as computing the full Hessian (https://github.com/google/jax/issues/3801). In our work, we show that the for a local output reconstruction objective, dropping of diagonal entries in Hessian can result in a suprisingly simple pruning metric. Our derivation also provides a new perspective as to how outliers features in LLMs might be preserved when pruning with SparseGPT. We hope these analysis results are useful for the pruning community.
>
> - **Is $|w|\cdot |x|$ the optimal pruning criterion for LLMs?**
> In Table 7, we have compared our proposed pruning metric $|w|\cdot|x|$ to existing pruning metrics in LLMs: magnitude $|w|$ and also the pruning metric of SparseGPT. We find that Wanda is optimal among the three pruning metrics. For the optimal comparison group, i.e. pruning per-output, the best pruned networks (perplexity 7.26) is obtained by our pruning metric.
>
> - **Latency speedup**
> We have evaluated the end to end latency speedup with structured sparsity. Specifically, we follow the setup of evaluating the speed of matrix multiplication. We observe a latency speedup of 1.24x on LLaMA-7B (structured 2:4 sparse: 251ms as compared to dense: 312ms). There is a gap to the matrix multiplication speedup on matrix multiplication (~1.6x). As suggested by the reviewer, we believe it is likely because there are operations other than fully connected layers in Transformers, e.g. attention computation. We have added this evaluation result in the updated draft.
>
> - **High sparsity**
> In our paper, we have provided the results on these two LLMs to give a fair and comprehensive evaluation of various pruning methods.  It is worth noting that for these two models, the standard magnitude pruning approach would give meaningless perplexity for OPT-13B (1e4) and Pythia-12B (3e5). Given that Wanda and magnitude pruning both do not involve weight update on the kept weights, our result provides a novel perspective that exact and effective sub-networks do exist for LLMs. In contrast, the prior work of SparseGPT shows that effective sub-networks exist in the neighborhood of the original weights. We hope these intriguing results could provide a new understanding on sparsity in LLMs.
>
> If you have other questions, we are happy to answer.
>
> [1] Importance estimation for neural network pruning. Molchanov, et al. CVPR 2019.
> [2] Optimal brain damage. Lecun et al, NeurIPS 1989.
> [3] SNIP: Single-shot Network Pruning based on Connection Sensitivity. Lee et al, ICLR 2019.

---

> > ### Comment · Reviewer_V88x · 2023-11-21
> > **Response to Rebuttal**
> >
> > Dear authors,
> >
> > Thanks for the comprehensive response.  I think the rebuttal addressed most of my questions. From my point of view, the 1.24$\times$ acceleration should not be considered a drawback, as I understand that the actual speed up is constrained by both hardware and software capabilities.
> >
> > But I still have one more question about the performance of 2:4 sparsity. Given that the latest PyTorch and Ampere GPU only support 2:4 semi-structured sparsity, the accuracy in this configuration is crucial for this paper. But the results for 2:4 sparsity presented in Tables 2 and 3 appear somewhat mixed, compared to SparseGPT.
> >
> > Besides, I noticed that D.4 involves fine-tuning the sparse model using LoRA. Could you provide more details about how to preserve the sparsity of the pruned model after merging the LoRA parameters into the model?

---

> > > ### Author Response · Authors · 2023-11-22
> > >
> > > We thank the reviewer for the response!
> > >
> > > **2:4 sparsity**
> > > We also observed this mixed result and discussed it in our paper. From our results, Wanda is more favorable for larger models, for example on LLaMA-30B and LLaMA-2-70B. While the trend is reversed for 2:4 sparsity on smaller models, we find that applying the weight update procedure on Wanda pruned weights can lead to better results than SparseGPT (see *Table 8 last column*). We have described this finding in the text.
> > >
> > > **LoRA fine-tuning**
> > > Merging the LoRA weights into the dense model will hurt the sparsity. In our experiment, we apply the LoRA adapters to the query and value projection matrices in the attention layers. The merging process won’t hurt the sparsity of unaffected layers, especially the computational intensive MLP layers with more parameters. For the LLaMA-1/2 family model, the overall sparsity after the weight merging would be 41.67%.
> > >
> > > We acknowledge that we do not explicitly consider the issue of sparsity loss when conducting the LoRA fine-tuning experiments. Our original goal was to explore and understand how fine-tuning could mitigate the performance drop of the pruning procedure. We adopted this fine-tuning strategy as it is a common method for fine-tuning LLM. To avoid any confusion, in the updated draft, we have moved this part to the section 5 and added a sentence to clarify our motivation.

---

### Official Review · Reviewer_JuRf · 2023-10-28

**Soundness:** 4 excellent
**Presentation:** 4 excellent
**Contribution:** 3 good
**Rating:** 6
**Confidence:** 4

**Summary:**

The paper proposes the implementation of the norm dot product of weights and activations (Wanda) as a pruning metric for LLMs. The primary motive is to preserve outlier weights within the LLM. Experiments were conducted on typical large models, such as Llama-v1 and Llama-v2. The results indicate that the proposed Wanda approach substantially reduces the time consumed by established pruning methods and enhances performance to a certain degree.

**Strengths:**

1. Although Wanda is quite simple and there are similar pruning metrics in the traditional pruning field, it is indeed buillt upon the consideration of outlier weights in LLM, rendering the narrative of the article easy to comprehend and follow. Thus, I believe that this paper holds significant value for the community.
2. The proposed Wanda method is highly efficient. Notably, it does not require backpropagation like SparseGPT, enhancing its applicability across various terminals to a considerable extent. Given that the performance of Wanda has essentially reached the optimal level, I deem it to have significant practical value.
3. The experiments are very thoroughness, making it very easy to understand the impact of various factors in LLM pruning, such as the pruning granularity and the effect of fine-tuning, among others. These elements also support my recommendation for the acceptance of this paper.

**Weaknesses:**

1. The authors primarily focus on experiments at a low sparsity rate of 50%, yet at a high sparsity rate (80%), Wanda's performance noticeably lags behind SparseGPT, which somewhat dampens my enthusiasm for this paper.
2. While the authors emphasize efficiency, and Wanda indeed greatly surpasses SparseGPT in efficiency (for example, 0.54s for WANDA and 203.1s for SparseGPT when pruning Llama-7b), I would like to question whether this disparity in time consumption truly holds value. My point being, the time expense for SparseGPT also remains quite low. Even though the authors claim the presence of certain real-time scenarios, I believe their notion of training sparse models from scratch is impractical, hence a more thorough discussion is warranted in this aspect.

**Questions:**

After fine-tuning with LoRA, the weights of LoRA will be incorporated into the pruned weights. As a result, the weights from LoRA will substantially mitigate the model's sparsity. Should the LoRA weights not be merged, however, additional inference time costs would ensue. Despite this not being the primary aspect of the paper, I maintain some reservations regarding this issue.

---

> ### Author Response · Authors · 2023-11-17
> **Response**
>
> We thank the reviewer for the positive assessment of our paper and the constructive comments. We are happy to address your concerns.
> - **High sparsity regime.**
> In our submitted version, we have provided the results on high sparsity to give a fair and comprehensive evaluation of various pruning methods. In the last part of section 5 (Table 8), we have analyzed why SparseGPT is superior over Wanda in this high sparsity regime. Specifically we show that the weight update procedure used by SparseGPT, i,e., reweighting scheme, is beneficial in high sparsity levels (e.g. 70%) but not on a lower sparsity level of 50%. We hope these intriguing results could provide deeper insights on these pruning methods.
>
> - **Pruning efficiency.**
> In recent years, training sparse models from scratch has been extensively studied as a research topic in network pruning. The lottery ticket hypothesis [1] articulates that there exists sparse networks that can be trained from scratch to match the performance of the dense model. As a followup, [2] is a representative work on sparse training that shows how to train high performance sparse networks from random initializations. In the sparse training techniques, pruning is applied iteratively throughout training: for example in [2], pruning is conducted every 100 iteration for 25k total iterations on ImageNet. Thus for such real-time applications, the pruning procedure needs to be extremely fast, e.g. done in seconds. In the conclusion of our submission, we have briefly discussed this point and various other related works on sparse training.
>
> - **LoRA fine-tuning**
> Merging the weights of LoRA adapters to the original model will make sparse weights lose the original sparsity level. However, the LoRA adapter is applied not on all the linear layers of the LLMs. In our experiments, we follow the standard LoRA procedure and adapt on the query and value projection matrix in the attention layers. In this case, the LoRA adapter would not affect the sparsity in the computational heavy MLP blocks.
>
>
>   We acknowledge that we do not consider this aspect of additional inference costs when doing the LoRA fine-tuning experiments. This is because our initial goal is to explore how fine-tuning could mitigate the performance drop of pruning. To avoid any confusion, in the updated draft, we have moved the part on fine-tuning to section 5 and have also added a sentence to clarify the motivation for this experiment.
>
> If you have other questions, we are happy to answer.
>
> [1] The Lottery Ticket Hypothesis: Finding Sparse, Trainable Neural Networks. Frankle et al, ICLR 2019.
> [2] Rigging the Lottery: Making All Tickets Winners. Evci et al, ICML 2020.

---

> > ### Comment · Reviewer_JuRf · 2023-11-21
> > **Official Comment by Reviewer JuRf**
> >
> > Thanks for the reply, which addressed my concerns. I will keep my positive rating.

---

> > > ### Author Response · Authors · 2023-11-21
> > >
> > > Thank you for the positive assessment and valuable feedback.

---

### Official Review · Reviewer_NmjJ · 2023-10-29

**Soundness:** 3 good
**Presentation:** 3 good
**Contribution:** 3 good
**Rating:** 5
**Confidence:** 4

**Summary:**

This work introduces a novel one-shot pruning approach for LLM, with its primary contribution being the introduction of a new weight importance score function. This function evaluates each weight by multiplying its magnitude with the norm of the corresponding input activations, which are estimated using a small set of calibration data. Extensive experiments on Llama and LLama2 demonstrate the effectiveness.

**Strengths:**

1. The work conducts extensive experiments and demonstrates that the pruned models outperform SparseGPT.
2. The method requires no retraining or weight update for LLMs, and the pruning speed is very fast (in seconds)

**Weaknesses:**

This work proposes a simple yet effective one-shot pruning method for LLMs, which has demonstrated superior performance over sparseGPT. However, I have concerns regarding its incremental contributions due to the following reasons:

1. The paper's introduction of a method to estimate weight importance based on both activation and weights does not appear to be novel. Similar concepts have been explored in previous works on LLM quantization, such as the AWQ work [1]
2. The pruning method proposed in the paper is simple and straightforward, which can be seen as an advantage. However, it operates under some strong assumptions, such as comparing and removing weights on a per-output basis, rather than adopting a more global pruning strategy. This could limit its applicability and effectiveness.
3. Given the limitation mentioned in point 2, although the method can be extended to semi-structured pruning, there are doubts about its ability to be extended to structured pruning.




[1] https://arxiv.org/abs/2306.00978

**Questions:**

Please refer to the weaknesses section.

---

> ### Author Response · Authors · 2023-11-17
> **Response**
>
> We thank the reviewer for the review and the constructive comments. We are happy to address your concerns.
> - **The paper's introduction of a method to estimate weight importance based on both activation and weights does not appear to be novel. Similar concepts have been explored in previous works on LLM quantization, such as the AWQ work [1].**
>
>   In our submitted version, we have cited the concurrent AWQ preprint. In LLM quantization, the idea of keeping outlier input  dimensions with large magnitude activations is not first proposed by AWQ. The prior published works of [1] and [2] (cited in our paper), both came out before AWQ and showed that input dimensions with large magnitudes are essential for the performance of LLM quantization, with AWQ doing W4 quantization and [1,2] doing W8A8 quantization. In our work, we demonstrate that these large magnitude outlier dimensions are important for network pruning.
>
> - **However, it operates under some strong assumptions, such as comparing and removing weights on a per-output basis, rather than adopting a more global pruning strategy. This could limit its applicability and effectiveness.**
>
>   For our method Wanda, we compare weights on a per-output basis. This is indeed a strong assumption, as we have discussed in Section 3 about “comparison groups”. However, we choose this comparison group because we empirically find that this leads to superior performances than other choices of comparison groups, e.g. layer-wise or comparing weights globally. In our paper (see Table 7, Table 12 and Table 13), we have shown that the per-output comparison group is superior over other comparison groups for a wide range of LLMs.
>
>
>   In our submission, we did not explore a more global pruning strategy. A global pruning strategy typically considers the objective functions when computing the pruning metrics. The pruning is done jointly with fine-tuning on a task-specific dataset, which is common in previous works on BERT pruning [3,4,5]. In the Appendix C of our paper (section *Additional Baselines*), we did a comparison of these pruning methods against our method Wanda, under the one-shot pruning setting followed by fine-tuning. We find that they are not able to find sparse networks in one shot.
>
>   Last, we would like to emphasize that our strong assumption on the comparison group does not affect its applicability to pruning metrics computed by a global pruning strategy. This is because the choice of pruning metric and the comparison group are orthogonal to each other, as a pruning metric and a comparison group jointly defines a pruning algorithm. One can search for the optimal pruning operation by experimenting different combinations of the pruning metric and the comparison group, as we did in Table 7.
>
> - **Given the limitation mentioned in point 2, although the method can be extended to semi-structured pruning, there are doubts about its ability to be extended to structured pruning.**
>
>   In our submission, we do not explore how our method Wanda can be extended to structured pruning. This is because the focus of this work is on weight-level pruning, which can be a much different pruning setting than structured pruning.  Our method Wanda is designed explicitly with that goal in mind: given a pre-trained dense LLMs, how to obtain a effective sparse LLM. Thus it might not be directly applicable to structured pruning.
>
>   Weight-level pruning and structured pruning have been two distinct sub-areas of network pruning. The differences between these two types of pruning method are discussed in the related work section of our submission. We have also cited a published work [6] on structured pruning of LLMs. We believe both research topics are as important for pruning LLMs and hope future works can benefit from this discussion.
>
> If you have other questions, we are happy to answer.
>
> [1] LLM.int8(): 8-bit Matrix Multiplication for Transformers at Scale. Dettmers et al, NeurIPS 2022.
> [2] SmoothQuant: Accurate and Efficient Post-Training Quantization for Large Language Models. Xiao et al, ICML 2023.
> [3] Movement pruning: Adaptive sparsity by fine-tuning. Sanh et al, 2020.
> [4] Platon: Pruning large transformer models with upper confidence bound of weight importance. Zhang et al, 2022.
> [5] Pruning Pre-trained Language Models with Principled Importance and Self-regularization. Ren et al, 2023.
> [6] LLM-Pruner: On the Structural Pruning of Large Language Models. Ma et al, NeurIPS 2023.

---

> > ### Comment · Reviewer_NmjJ · 2023-11-22
> > **Response to rebuttal**
> >
> > Thank you for your response. My main concern remains the technical novelty and contribution of the paper. I did not mean to imply that you did not cite AWQ, but rather that the idea of estimating weight importance based on both activation and weights has been widely used in quantization work. Wanda may be the first to apply this idea to LLM pruning's unstructured and n:m sparsity. Based on this, I expect Wanda's application scope to be broader, such as generalizing to the range of structured pruning. However, the author's response mentioned that Wanda does not apply in structured pruning.

---

> ### Author Response · Authors · 2023-11-22
>
> We thank the reviewer for the response!
>
> **The idea of estimating weight importance based on both activation and weights has been widely used in quantization work.**
> In W8A8 quantization, the prior works of LLM.int8() and SmoothQuant focus on protecting the outlier input channels with large magnitudes. There is *rare* or *little* notion of the concept *weight importance* in these methods as their core ideas are to protect these outlier input channels from activation quantization. To the best of our knowledge, the use of the concept *weight importance* in quantization is mostly brought up by AWQ . Specifically they use both weight and activation information to identify the important weights critical for the quantization performance.
>
> Could the reviewer provide some references on previous quantization works that estimate weight importance based on both the weights and activations? We would be happy to cite and discuss them in the next revision.
>
> **Wanda does not apply in structured pruning.**
> We acknowledge that the scope of this paper is focused on unstructured and semi-structured sparsity. We will clarify this in the next revision. The focus on unstructured/semi-structured pruning in our experiments is mainly to follow existing works [1,2] in this area.
>
> [1] The Lottery Ticket Hypothesis: Finding Sparse, Trainable Neural Networks. 2019.
> [2] SparseGPT: Massive Language Models Can Be Accurately Pruned in One-Shot. 2023.

---

### Official Review · Reviewer_tLFR · 2023-11-01

**Soundness:** 4 excellent
**Presentation:** 4 excellent
**Contribution:** 3 good
**Rating:** 8
**Confidence:** 4

**Summary:**

The manuscript shows structured and unstructured pruning of LLMs can be successful if the pruning score accounts for the magnitudes of the $\textbf{W}$eight $\textbf{and}$ the $\textbf{a}$ctivation that the weight multiplies ($\textbf{Wanda}$). This approach is motivated by the observation that outlier activations with large magnitudes exist in LLMs, so pruning based on weight magnitudes alone can destroy salient computations of LLMs.

The pruning approach requires one forward pass, does not necessitate retraining, and causes minor performance degradation when applied to pretrained networks (larger networks are more tolerant of Wanda and pruning generally).

The reasons for Wanda's strong performance (e.g., against competitors like SparseGPT) are explored in convincing mathematical analyses and ablation studies that overall support the design decisions of Wanda, despite their simplicity.

**Strengths:**

Broadly, the manuscript gives timely insight and intuition for the problem of LLM pruning. Its proposed approach solves several problems with LLM pruning, making it faster, more performant, and simpler.

The authors make a helpful connection of their approach (Wanda) to existing work (SparseGPT), showing the similarity of their pruning scores when an assumption is made on the Hessian structure. This helps justify the pruning score used by Wanda, which is surprisingly principled given its simplicity (the connection to LeCun et al., 1989, is also nice).

The observation that Wanda's choice of granularity ("[comparing] and [removing] weights on a per-output basis") is important to LLM performance but not image classifier performance is very significant and (as far as I know) original.

**Weaknesses:**

The main weakness of the manuscript is that it leaves unclear the benefits of Wanda to inference speed. While Wanda can accelerate matrix multiplications (Table 6), readers will be left curious about how inference timings are affected by Wanda. As my question below clarifies, this can be easily addressed.

**Questions:**

Score-affecting:

1. Can you please show full-model inference speeds for Wanda-pruned and dense networks? If you contextualize the matrix multiplication speedups of Table 6 with inference timings, the efficiency community can better understand the most relevant ways to build on the submitted work.
   - For example, as a reader interested in speeding up inference, I am unsure if the most promising next step is to figure out how to prune more weights with Wanda, or if the best next step is finding a distinct inference-speedup technique that complements Wanda's speedup of matrix multiplication.

Minor:

1. I believe Wanda prunes the weights of a given layer before computing the activations that will be used to prune the next layer, and the manuscript supports this by saying "After pruning a preceding layer, the subsequent layer receives updated input activations, based on which its pruning metric will be computed". I would suggest rephrasing this statement slightly to make it clear that the the "updated input activations" are those created by the pruned weights of the prior layer.

2. Is this really an "element wise dot product" (page 3), and not a Hadamard product?

3. On page 4, $\mathrm{diag}$ is used twice, once on a matrix and once on a scalar. I think the usage on the scalar is conveying that we have a diagonal matrix with each entry being the squared $\ell_2$ norm of the $j$th feature. If that's right, please consider if you can improve this sentence's notation to avoid confusion (e.g., change use of $\mathrm{diag}$ or its argument).

---

> ### Author Response · Authors · 2023-11-17
> **Response**
>
> We thank the reviewer for the positive assessment of our paper and the constructive comments. We are happy to address your concerns.
> - **Full-model inference speedup**
>
> For end to end latency speedup with structured sparsity, we follow the setup of evaluating matrix multiplication.  We observe a latency speedup of 1.24x on LLaMA-7B (structured 2:4 sparse: 251ms as compared to dense: 312ms). There is indeed a gap to the speedup purely on matrix multiplication (~1.6x). This is likely because there are operations other than fully connected layers in Transformers, e.g. attention computation. We have updated the draft with this evaluation result.
>
> - **Prune more weights, or find complementary inference-speedup techniques**
>
> The speedup experiment in our submission is evaluated on structured 2:4 sparsity, where the sparsity is fixed to be 50%. For semi-structured sparsity, it is likely that we can not gain more speedup by pruning more weights over 50%. For unstructured sparsity, there might be room for inference improvement with higher sparsity.
>
> Our results on the full-model inference speedup suggest that finding complementary inference-speedup techniques might be a promising future direction. For example, techniques for exploiting sparsity in attention [1,2] are orthogonal to structured sparsity. Another example would be to combine structured sparsity with quantization methods.
>
> - **Rephrase the statement slightly to make it clear that the "updated input activations" are those created by the pruned weights of the prior layer.**
>
> In the updated submission, we have revised this sentence to make it clear that we are using the updated activations computed by the pruned weights of the previous layer. The new sentence is as follows:
> *After pruning a preceding layer, the current layer receives the updated input activations, obtained on the pruned weights of the previous layer.*
>
> - **Is this an "element wise dot product" (page 3), and not a Hadamard product?**
>
> The Hadamard product [3] takes in two matrices of the same dimensions and returns a matrix of the elementwise multiplied elements. In equation 1, our weight importance is computed as a product of two scalar values. Therefore, it is not a Hadamard product. Specifically, the weight $W$ and input activation norms $||X||$ do not have the same shape: $W$ is of shape $(C_{\text{out}}, C_{\text{in}})$ and $||X||$ is a vector of shape $(C_{\text{in}},)$.
>
> In the caption of Figure 1, we have indeed used the term “elementwise product” to describe how we compute the weight importance. To avoid the confusion with Hadamard product, we have removed the use of the word “elementwise” in the updated version.
>
> - **Improve this sentence's notation to avoid confusion (e.g., change use of $\text{diag}$ or its argument)**
>
> We have updated this part with two modifications:
> 1.We now avoid the use of $\text{diag}$ operator on the scalar but instead refer to it as “the $j$th diagonal of $X^{T}X$”. Now the $\text{diag}$ operator is only applied to the matrix.
> 2.We have added an additional equation to show the exact reduction steps, which should make this part more clear, especially the use of $\text{diag}$ on the matrix.
>
> If you have other questions, we are happy to answer.
>
> [1] Deja vu: Contextual sparsity for efficient LLMs at inference time. Liu et al, ICML 2023.
> [2] H2O: Heavy-Hitter Oracle for Efficient Generative Inference of Large Language Models. Zhang et al, NeurIPS 2023.
> [3] https://en.wikipedia.org/wiki/Hadamard_product_(matrices)

---

> > ### Comment · Reviewer_tLFR · 2023-11-20
> >
> > My concerns have been addressed. Thank you for the new results!
> >
> > By the way, I was questioning the use of "dot product", not the use of "element wise".

---

> > > ### Author Response · Authors · 2023-11-21
> > >
> > > Thank you for the positive assessment and valuable feedback.
> > >
> > > Here the use of “dot product” is applied on two scalar variable values, which we will make this point more clear in the final version.  From the original review, we felt the use of “element-wise” was also an issue, which might be confused with the definition of Hadamard product.

---

### Meta-Review · Area_Chair_QGh1 · 2023-12-10

**Metareview:**

Reviewers found this work solid and of practical interest. The claims are supported and the rebuttal was crisp and effective, clarifying all the reviewers' concerns. I would encourage the authors to incorporate the discussion points and new results they provided during the rebuttal. One reviewer mentioned that similar ideas have appeared in the literature, referring to AWQ (https://arxiv.org/abs/2306.00978). While this work is indeed similar, it is a pre-print that is concurrent with the current work. The authors are citing this work in the revised version of the paper, which is satisfactory.

**Justification For Why Not Higher Score:**

While this is solid work and well executed, there are no remarkable results or contributions that would call for a spotlight (or oral).

**Justification For Why Not Lower Score:**

All reviewers agreed this was a solid paper that should be accepted.

---

### Decision · Program_Chairs · 2024-01-16

Accept (poster)